# Targeting the IRE1α/XBP1s pathway suppresses CARM1-expressing ovarian cancer

Jianhuang Lin [1], Heng Liu[1], Takeshi Fukumoto[1], Joseph Zundell[1], Qingqing Yan[2], Chih-Hang Anthony Tang[3], Shuai Wu [1], Wei Zhou[1], Dajiang Guo [1], Sergey Karakashev[1], Chih-Chi Andrew Hu [3], Kavitha Sarma[2], Andrew V. Kossenkov[2] & Rugang Zhang [1]✉

CARM1 is often overexpressed in human cancers including in ovarian cancer. However, therapeutic approaches based on CARM1 expression remain to be an unmet need. Cancer cells exploit adaptive responses such as the endoplasmic reticulum (ER) stress response for their survival through activating pathways such as the IRE1α/XBP1s pathway. Here, we report that CARM1-expressing ovarian cancer cells are selectively sensitive to inhibition of the IRE1α/XBP1s pathway. CARM1 regulates XBP1s target gene expression and directly interacts with XBP1s during ER stress response. Inhibition of the IRE1α/XBP1s pathway was effective against ovarian cancer in a CARM1-dependent manner both in vitro and in vivo in orthotopic and patient-derived xenograft models. In addition, IRE1α inhibitor B-I09 synergizes with immune checkpoint blockade anti-PD1 antibody in an immunocompetent CARM1-expressing ovarian cancer model. Our data show that pharmacological inhibition of the IRE1α/XBP1s pathway alone or in combination with immune checkpoint blockade represents a therapeutic strategy for CARM1-expressing cancers.

[1] Immunology, Microenvironment and Metastasis Program, The Wistar Institute, Philadelphia, PA, USA. [2] Gene Expression and Regulation Program, The Wistar Institute, Philadelphia, PA, USA. [3] Center for Translational Research in Hematologic Malignancies, Houston Methodist Cancer Center, Houston Methodist Research Institute, Houston, TX, USA. ✉email: rzhang@wistar.org

The development of therapeutic strategies for epithelial ovarian cancer (EOC) remains a major obstacle to overcome. High-grade serous ovarian cancer (HGSOC) is the most common and fatal subtype of EOC. EOC is genetically heterogeneous[1]. Thus, it is imperative that therapeutic strategies need to be personalized by targeting distinct molecular subsets of EOC[2]. CARM1 (also known as PRMT4) is a type I protein arginine methyltransferase (PRMT) that asymmetrically dimethylates arginine residues on protein substrates[3,4]. Overwhelming evidence suggests that CARM1 functions as an oncogene in human cancers[5]. High levels of CARM1 expression have been observed in several major cancer types, including breast, colon, and prostate[6–8]. Notably, CARM1 amplification/overexpression occurs in ~20% of HGSOCs[9], the highest among all cancer types in The Cancer Genome Atlas (TCGA) databases. However, the role of CARM1 and the associated therapeutic vulnerabilities conferred by its expression in cancers including EOC remain to be explored.

The endoplasmic reticulum (ER) stress response or the unfolded protein response (UPR) orchestrates adaptive programs to promote cancer cell survival[10–12]. Thus, inhibition of the UPR represents a therapeutic approach for cancers with hyperactive ER stress response[12]. The mammalian UPR is governed by three stress transducers that sense ER stress. They include inositol-required enzyme 1 alpha (IRE1α), activating transcription factor 6 (ATF6), and protein kinase RNA-like ER kinase (PERK)[12]. When functions of the ER are severely impaired, cells undergo apoptosis through effector proteins such as C/EBP homologous protein (CHOP) downstream of PERK[13]. IRE1α signaling is the most conserved and well-studied UPR. In response to ER stress, the endoribonuclease activity of the IRE1α RNase domain is activated by conformational changes[12]. IRE1α RNase processes the mRNA encoding the transcription factor X-box binding protein 1 (XBP1), excising a 26-nucleotide intron in the XBP1 mRNA[12]. This splicing event shifts the coding reading frame, leading to the translation of a transcription factor termed spliced XBP1 (XBP1s). XBP1s translocates into the nucleus to promote the transcription of genes involved in protein folding among other targets to alleviate ER stress[12]. However, the mechanism that controls the IRE1α/XBP1s pathway remains poorly understood. For example, how XBP1s target genes were regulated at the molecular and genome-wide levels is not clear.

In addition to tumor-intrinsic function, the ER stress response such as the IRE1α/XBP1s pathway is implicated in intratumoral immune cells[10,11]. For example, activation of the IRE1α/XBP1s pathway is known to be immune suppressive in various populations of immune cells[10]. Targeting the ER stress response may reinvigorate endogenous antitumor immunity, which could synergize with immunotherapies, such as immune checkpoint blockade[10]. Thus, there are substantial ongoing efforts in exploring the pathway as a cancer therapeutic target. However, the molecular determinant of therapeutic response to the inhibition of the IRE1α/XBP1s pathway remains poorly understood.

Here we show that pharmacological targeting of the IRE1α/XBP1s pathway selectively suppresses CARM1-expressing ovarian cancer, which further synergizes with immune checkpoint blockade. Mechanistically, CARM1 determines ER stress response through controlling the IRE1α/XBP1s pathway by forming a complex with XBP1s to regulate its target gene expression.

## Results

**CARM1 determines XBP1s target gene expression.** To systematically profile CARM1 distribution pattern genome-wide, we performed the cut-and-run analysis for CARM1 in CARM1-expressing A1847 HGSOC cells[9]. The analysis revealed 22,398 significant peaks (false discovery rate (FDR) < 5% with at least fourfold over input). Overall distribution of CARM1 peak signal coincided with active histone mark H3K27ac binding and proximity to transcription starting sites (TSS) (Fig. 1a), which is consistent with the reported role of CARM1 as a transcriptional activator[4]. Notably, 25% of all CARM1-binding sites overlapped with genes' TSS with 7859 genes occupied by CARM1 at the promoter (Fig. 1b). Analysis of enrichment of known regulators for those genes revealed MYC as the top regulator, which is consistent with previous reports that CARM1 regulates the c-MYC pathway[3]. Notably, XBP1s was identified as a top predicted regulator of the identified putative CARM1 target genes (Fig. 1c). Indeed, de novo motif analysis of 100 bp regions around TSS of CARM1-bound genes by *HOMER* revealed a significant enrichment of ACGTCA motif ($P = 10^{-101}$) matching the core of known XBP1s-binding motif (Fig. 1d). This raised the possibility that CARM1 may regulate XBP1s function during ER stress response. To test this possibility, we performed XBP1s and CARM1 cut-and-run analysis with or without ER stress-inducer tunicamycin treatment (Supplementary Fig. 1a). The analysis revealed that CARM1- and XBP1s-binding signal significantly correlated in tunicamycin treatment condition and both factors increased binding compared to vehicle control-treated cells (Fig. 1e–g). Thus, we conclude that CARM1 is associated with XBP1s target genes.

We next sought to identify direct CARM1 and XBP1s target genes that are subjected to ER stress regulation. Toward this goal, we performed RNA sequencing (RNA-seq) in control, CARM1 knockout, and XBP1s knockdown A1847 cells treated with or without tunicamycin (Fig. 2a, b). The analysis revealed that 3313 genes were upregulated by tunicamycin in A1847 cells. XBP1s knockdown significantly downregulated response of 779 of the 3313 genes induced by tunicamycin. In addition, CARM1 knockout significantly downregulated response of 1722 of the 3313 genes induced by tunicamycin. Five hundred and forty-three of the 3313 genes were downregulated by both XBP1s knockdown and CARM1 knockout, which represents a statistically significant overlap ($P < 10^{-10}$ by hypergeometric test; Fig. 2c, d). We next cross-referenced the 543 genes with XBP1s and CARM1 peaks identified in A1847 cells treated with ER stress-inducer tunicamycin to identify direct ER stress-induced XBP1s and CARM1 target genes. The analysis revealed that 363 of the 543 genes are directly associated with both XBP1s and CARM1 in their promoters ($P < 10^{-10}$ by hypergeometric test; Fig. 2e, f and Supplementary Data 1). In addition, we observed a significant correlation between strength of XBP1s binding and the increase in CARM1 binding under ER stress condition induced by tunicamycin (Fig. 2g). These findings suggest that CARM1 regulates XBP1s's association with its target genes during ER stress response.

We next sought to correlate expression of CARM1 with identified CARM1/XBP1s target genes under normal condition. Toward this goal, we cross-referenced RNA-seq data of control A1847 cells and CARM1 knockout or XBP1 knockdown A1847 cells to identify genes commonly regulated by CARM1 and XBP1s without ER stress induction. In addition, we examined the binding of both CARM1 and XBP1s on these genes based on the cut-and-run datasets. The analysis revealed 430 CARM1/XBP1s direct target genes that are downregulated by CARM1 knockout or XBP1 knockdown at the basal level (Supplementary Fig. 2a and Supplementary Data 2). We explored the correlation between CARM1 and the identified CARM1/XBP1s target genes in the TCGA HGSOC dataset. Indeed, there is a significant correlation between expression of CARM1 and these genes in the TCGA HGSOC dataset (Fig. 2h). A similar positive correlation was also obtained in the Broad Institute Cancer Cell Line Encyclopedia database (Supplementary Fig. 2b).

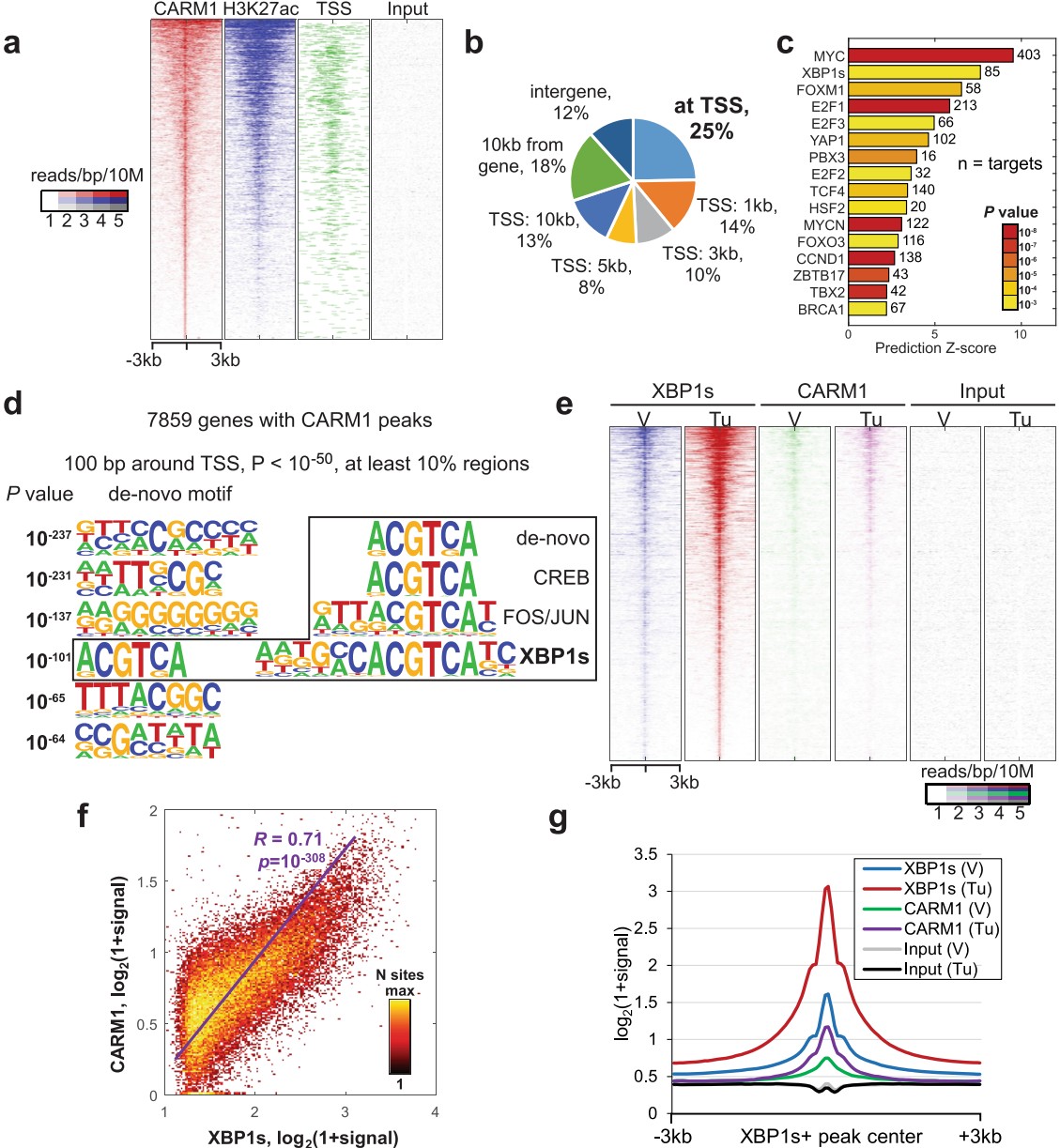

**Fig. 1 CARM1-binding sites are enriched for XBP1s consensus. a** Heatmap of cut-and-run profiles of CARM1, H3K27ac, and input signal. Binding sites are sorted by strength of CARM1 signal. 3 kb around CARM1 peak centers are shown with 100 60 bp bins. **b** Genomic-wide distribution of CARM1 peaks relative to gene. **c** Ingenuity Pathway analysis of known regulators enriched among 7859 genes bound by CARM1. Transcriptional regulators with activation prediction $Z$-score of at least 2 and $P < 0.001$ are shown. The $P$ value in Ingenuity Pathway analysis measures whether there is a statistically significant overlap between dataset genes and the genes that are regulated by a regulator. It is calculated using two-tailed Fisher's Exact Test. **d** De novo motif analysis of 100 bp TSS regions for 7859 genes bound by CARM1. **e** Heatmap of cut-and-run profiles of XBP1s, CARM1, and input signal for vehicle (V) or ER stress-inducer tunicamycin (Tu)-treated cells around center of XBP1s peaks. Binding sites are sorted by strength of XBP1s signal. 3 kb around XBP1 peak centers are shown with 100 60 bp bins. **f** Correlation of XBP1s and CARM1 binding signal in XBP1s peak regions in tunicamycin-treated cells. $P$ value was calculated using two-tailed test of significance. **g** Average profiles of binding signals for the indicated samples as shown in **f**.

We validated the downregulation of the identified CARM1/XBP1s genes such as *HSPA5* and *DNAJB9* by CARM1 knockout using quantitative reverse-transcriptase (qRT-PCR) analysis in A1847 cells (Supplementary Fig. 2c). Similar downregulation of CARM1/XBP1s target genes by CARM1 knockout was also observed in CARM1-expressing PEO4 and OVSAHO cells (Supplementary Fig. 2d–g). Conversely, we sought to determine whether upregulation of CARM1 in CARM1-low cells upregulates CARM1/XBP1s target genes. Toward this goal, we specifically upregulated endogenous CARM1 expression using a CRISPR (clustered regularly interspaced short palindromic

repeats)-mediated activation system (Fig. 2i)[14]. Indeed, CARM1 activation promoted CARM1/XBP1s target gene expression in both EF027 and COV362 cells (Supplementary Fig. 2h). These findings suggest that the observed effects are not cell line specific. Notably, the upregulation of these genes triggered by ER stress-inducer tunicamycin was also suppressed by CARM1 knockout (Fig. 2j and Supplementary Fig. 2i). Consistently, XBP1s reporter activity was significantly lower in CARM1 knockout cells compared with controls in response to tunicamycin treatment (Fig. 2k). In contrast, XBP1s reporter activity was significantly increased by CARM1 activation in tunicamycin-treated

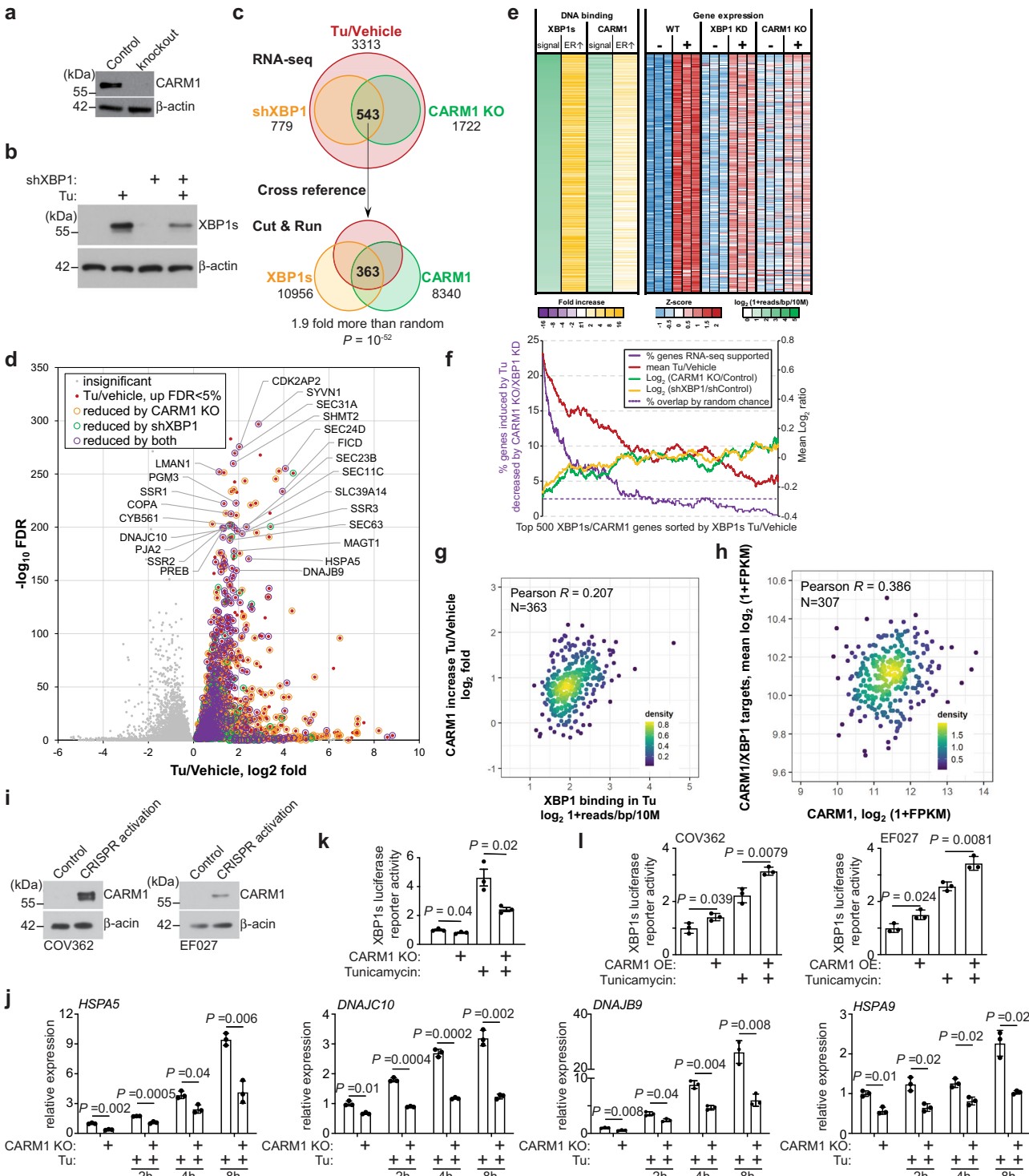

cells (Fig. 2l). Together, these results show that CARM1 determines the expression of XBP1s target genes during ER stress response.

We next validated the binding of both CARM1 and XBP1s with CARM1/XBP1s target genes such as *HSPA5* and *DNAJB9* by chromatin immunoprecipitation (ChIP) analysis (Fig. 3a, b). Notably, CARM1 knockout significantly impaired the recruitment of XBP1s to these genes (Fig. 3b), which is consistent with the observed downregulation of CARM1/XBP1s target genes in CARM1 knockout cells (Fig. 2g). As a control, the level of H3R17me2a, a product of the CARM1's enzymatic activity[4], on

the promoters of these genes was also decreased by CARM1 knockout (Fig. 3b). Conversely, we sought to determine whether XBP1s affects CARM1 recruitment to CARM1/XBP1s target genes during ER stress response. Indeed, XBP1 knockdown significantly impaired the recruitment of CARM1 to the promoters of the CARM1/XBP1s target genes, which correlated with a decrease in H3R17me2a on these genes' promoters (Fig. 3c). Finally, sequential ChIP analysis using an anti-CARM1 antibody followed by an anti-XBP1 antibody showed a significant enrichment of CARM1/XBP1s on these genes' promoters (Fig. 3d). Together, we conclude that XBP1s and CARM1 are

**Fig. 2 CARM1 promotes the expression of XBP1s target genes. a** Expression of CARM1 and a loading control β-actin in control and CARM1 knockout (KO) A1847 cells determined by immunoblot. Immunoblots are representative of three biologically independent experiments with similar results. **b** Expression of spliced XBP1 (XBP1s) and a loading control β-actin in vehicle control or tunicamycin (5 μg/ml, 8 h) treated A1847 cells determined by immunoblot. Immunoblots are representative of three biologically independent experiments with similar results. **c** Significant overlap of XBP1s and CARM1-regulated genes during ER stress response induced by tunicamycin. Overlap of tunicamycin-induced genes (Tu/Vehicle) whose response was downregulated by XBP1 knockdown (shXBP1) or CARM1 knockout (CARM1 KO) determined by RNA-seq revealed a list of 543 genes. Cross-referencing the 543 genes with XBP1s and CARM1 cut-and-run identified a list of 363 direct XBP1s/CARM1 target genes in response to tunicamycin treatment. P values were calculated by two-tailed Fisher's Exact Test. **d** Volcano plot for genes induced by tunicamycin determined by RNA-seq analysis. Tunicamycin induced genes that are regulated by CARM1, XBP1s, or both are indicated by different dot shapes. The genes are regulated by both CARM1 and XBP1s are highlighted. **e** Heatmap of cut-and-run and RNA-seq analysis for the identified 363 CARM1/XBP1s direct target genes. **f** Correlation between CUT&RUN and differential gene expression using 500 top XBP1s peak-mapped genes. CUT&RUN peak-mapped genes are sorted by XBP1s-binding fold change over vehicle. Using a 500-gene window, 4 values were plotted: percentage of genes that are induced by tunicamycin and decreased by both CARM1 KO and shXBP1, mean $\log_2$ ratio of Tu/vehicle, mean $\log_2$ (CARM1 KO/Ctrl), and mean $\log_2$ (shXBP1/Ctrl). **g** Heatmap of cut-and-run and RNA-seq data for the identified 363 CARM1/XBP1s direct target genes. **h** Correlation between tunicamycin-induced increase of CARM1 binding and strength of XBP1 binding at the 363 CARM1/XBP1s direct target genes. **i** Correlation between the expression of CARM1 and CARM1/XBP1s direct target genes identified at the basal level in the high-grade serous ovarian cancer (HGSOC) TCGA dataset. **j** Expression of CARM1 and a loading control β-actin in control and CRISPR-mediated activation of endogenous CARM1 expressing COV362 and EF027 cells determined by immunoblot. Immunoblots are representative of two biologically independent experiments with similar results. **k** RT-qPCR results showing the expression of the indicated CARM1/XBP1s target genes in control and CARM1 knockout A1847 cells treated with vehicle control or tunicamycin (5μg/ml, 8 h). **l** XBP1s luciferase reporter activity in control and CARM1 knockout A1847 cells treated with vehicle control or tunicamycin (5 μg/ml, 16 h). **l** XBP1s luciferase reporter activity in control and CARM1 overexpressed (OE) COV263 and EF027 cells treated with vehicle control or tunicamycin (5 μg/ml, 16 h). Data represent mean ± SEM, n = 3 biologically independent experiments unless otherwise stated. P values were calculated using two-tailed *t* test except in **c** by Fisher's Exact Test and in **g**, **h** by Pearson R analysis.

mutually dependent upon each other for their optimal recruitment to the CARM1/XBP1s target genes.

**CARM1 and XBP1s interact with each other.** Given the observed mutual dependence of CARM1 and XBP1s for their optimal recruitment to the CARM1/XBP1s target genes in response to ER stress, we next determined whether CARM1 and XBP1s interact with each other. Indeed, co-IP revealed an interaction between CARM1 and XBP1s in response to tunicamycin treatment (Fig. 4a). The observed interaction was validated by glutathione *S*-transferase (GST) pulldown using GST-tagged CARM1 in lysates of A1847 cells treated with tunicamycin (Fig. 4b). We next sought to examine the interaction between XBP1s and CARM1 in situ by proximity ligation assay (PLA) in cells treated with or without tunicamycin. While we observed a basal level of interaction between XBP1s and CARM1 in control cells compared with negative immunoglobulin G (IgG) control, the interaction was significantly enhanced by ER stress-inducer tunicamycin (Fig. 4c, d). Together, we conclude that ER stress enhances CARM1 and XBP1s interaction, which correlates with the induced expression of XBP1s.

We next sought to map the CARM1 domain that interacts with XBP1s. GST-pulldown analysis of GST-tagged control or mutant CARM1 revealed that the catalytic domain of CARM1 (141–480) interacts with XBP1s in response to ER stress (Supplementary Fig. 3a, b). Indeed, ectopic CARM1 141–480 expression suppressed the expression of CARM1/XBP1s target genes (Fig. 4e, f and Supplementary Fig. 3c). This result suggests that the catalytic domain of CARM1 may function as a dominant-negative mutant in regulating CARM1/XBP1s target gene expression. Consistently, inhibition of CARM1 enzymatic activity does not affect the expression of CARM1/XBP1s target genes (Supplementary Fig. 3d, e). Notably, inhibition of CARM1 enzymatic activity does not affect the inhibition of IRE1α activity by B-I09 as determined by the ratio between total and spliced *XBP1* mRNAs (Supplementary Fig. 3f). This result indicates that the catalytic activity of CARM1 is not required for the observed phenotype and instead the catalytic domain serves as an interaction domain with XBP1s. Together, we conclude that the catalytic domain of CARM1 mediates its interaction with XBP1s in an enzymatic activity-independent manner to regulate CARM1/XBP1s target gene expression.

**CARM1 determines response to inhibition of the IRE1α/XBP1s pathway.** Since we show that CARM1 promotes the IRE1α/XBP1s pathway, we next determined whether inhibition of the IRE1α/XBP1s pathway is selective against CARM1 expression. Toward this goal, we treated control and CARM1 knockout A1847 cells with a selective IRE1α inhibitor B-I09[15]. We chose B-I09 for our study because of its ability to specifically target IRE1α RNase activity and safety profile in vivo in preclinical studies[15–19]. Indeed, compared with controls, CARM1 knockout increased the $IC_{50}$ of B-I09 in A1847 cells (Fig. 5a, b). Likewise, CARM1 knockout reduced the growth rate inhibition by B-I09 (Supplementary Fig. 4a, b)[20]. Consistent with the notion that unresolved ER stress leads to apoptosis, B-I09 treatment upregulated apoptotic markers such as cleaved lamin A and cleaved poly ADP-ribose polymerase (PARP) p85 in A1847 cells (Fig. 5c). As a control, B-I09 failed to induce apoptotic markers in CARM1 knockout A1847 cells (Fig. 5c). Similar increase in $IC_{50}$ was also observed for another IRE1α inhibitor 4μ8c (Supplementary Fig. 4c). Likewise, CARM1 knockout increased the $IC_{50}$ of IRE1α inhibitor in PEO4 and OVSAHO cells (Supplementary Fig. 4d–f). Conversely, endogenous CARM1 upregulation decreased the $IC_{50}$ of B-I09 in CARM1-low cell lines (Fig. 5d). Notably, B-I09 is equally effective in suppressing IRE1α enzymatic activity in these cells as determined by the ratio between *XBP1s* and *XBP1* in these cells (Supplementary Fig. 4g). Finally, we determined the $IC_{50}$ of B-I09 in a panel of HGSOC cell lines. The $IC_{50}$ of B-I09 was significantly lower in CARM1-high cell lines compared with those cell lines with low CARM1 expression (Fig. 5e–g). Consistently, in the Project Achilles synthetic lethality database[21], *XBP1* short hairpin RNA (shRNA) was more effective in suppressing the growth of cell lines with high CARM1 expression compared with those with low CARM1 expression (Supplementary Fig. 4h, i). Notably, the observed differences in response to B-I09 in the panel of cell lines did not correlate with c-MYC expression (Fig. 5e). Consistently, CARM1 knockout did not affect c-MYC expression (Supplementary Fig. 4j). Notably, inhibition of the enzymatic activity of CARM1 did not affect the sensitivity to B-I09 (Supplementary Fig. 4k), which is consistent with the finding that inhibition of CARM1 enzymatic activity does not affect the expression of CARM1/XBP1s target genes (Supplementary Fig. 3d, e). In addition, the observed effects were not due

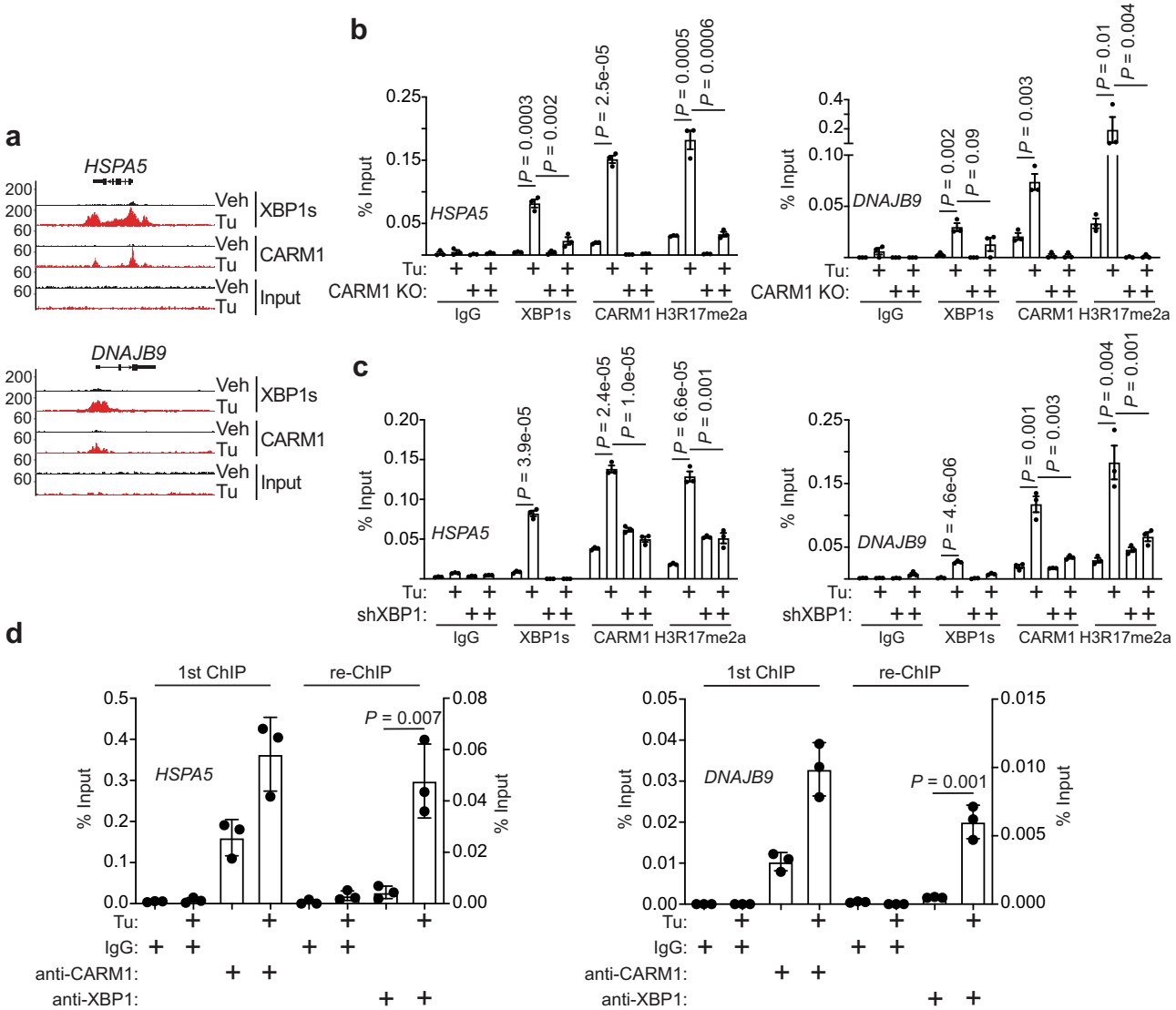

**Fig. 3 CARM1 and XBP1s are mutually dependent for their optimal recruitment to CARM1/XBP1s target genes. a** Representative cut-and-run tracks of CARM1 and XBP1s on the indicated CARM1/XBP1s target *HSPA5* and *DNAJB9* gene loci in cells treated with vehicle control or tunicamycin (5 μg/ml, 8 h). **b** ChIP-qPCR analysis for the association of XBP1s, CARM1, and H3R17me2a, an enzymatic product of CARM1, with the promoters of *HSPA5* and *DNAJB9* genes in control or CARM1 knockout A1847 cells treated with vehicle control or tunicamycin (5 μg/ml, 8 h). **c** ChIP-qPCR analysis for the association of XBP1s, CARM1, and H3R17me2a with the promoters of *HSPA5* and *DNAJB9* genes in control or XBP1 knockdown A1847 cells treated with vehicle control or tunicamycin (5 μg/ml, 8 h). **d** Sequential ChIP-qPCR analysis using an anti-CARM1 antibody (first ChIP) followed by an anti-XBP1 antibody (re-ChIP) for the promoters of *HSPA5* and *DNAJB9* genes in A1847 cells. Data represent mean ± SEM, *n* = 3 biologically independent experiments. *P* values were calculated using two-tailed *t* test.

to variation in the expression of *XBP1*, *XBP1s*, or the ratio between *XBP1s* and *XBP1* among these cell lines (Supplementary Fig. 4l). Likewise, CARM1 knockout did not affect XBP1s induction in response to tunicamycin treatment (Supplementary Fig. 4m). Conversely, CARM1 activation did not affect either IRE1α expression or XBP1s induction in response to tunicamycin (Supplementary Fig. 4n). These findings support the notion that CARM1-regulated recruitment of XBP1s to its target genes instead of its expression dictates the response to IRE1α inhibitor B-I09. Together, we conclude that inhibition of the IRE1α/XBP1s pathway is selective against CARM1 expression in HGSOC cells.

Consistent with findings that B-I09 induces apoptosis in CARM1-expressing cells, the pro-apoptotic ER stress effector CHOP expression[22] was induced by B-I09 in CARM1-expressing cells (Supplementary Fig. 4o, p). Indeed, CHOP knockdown significantly decreased the sensitivity of B-I09 in CARM-

expressing cells (Fig. 5h and Supplementary Fig. 4q), which correlates with the suppression of markers of apoptosis (Fig. 5i). These results support that CARM1 coordinates pro-apoptotic vs. pro-survival ER stress response and that CHOP signaling contributes to apoptosis induced by B-I09 in CARM1-expressing cells.

**IRE1α inhibition suppresses the growth of CARM-high EOCs in vivo**. To determine the effects of IRE1α inhibition on CARM1-high EOCs in vivo, we utilized three different models. In the orthotopic model, the tumors established by CARM1-expressing A1847 cells in mouse bursa that covers mouse ovary were treated with vehicle control or B-I09 for 2 weeks (Supplementary Fig. 5a). At the end of treatment, we used tumor weight as a surrogate for tumor burden. Compared with vehicle control, B-I09 significantly

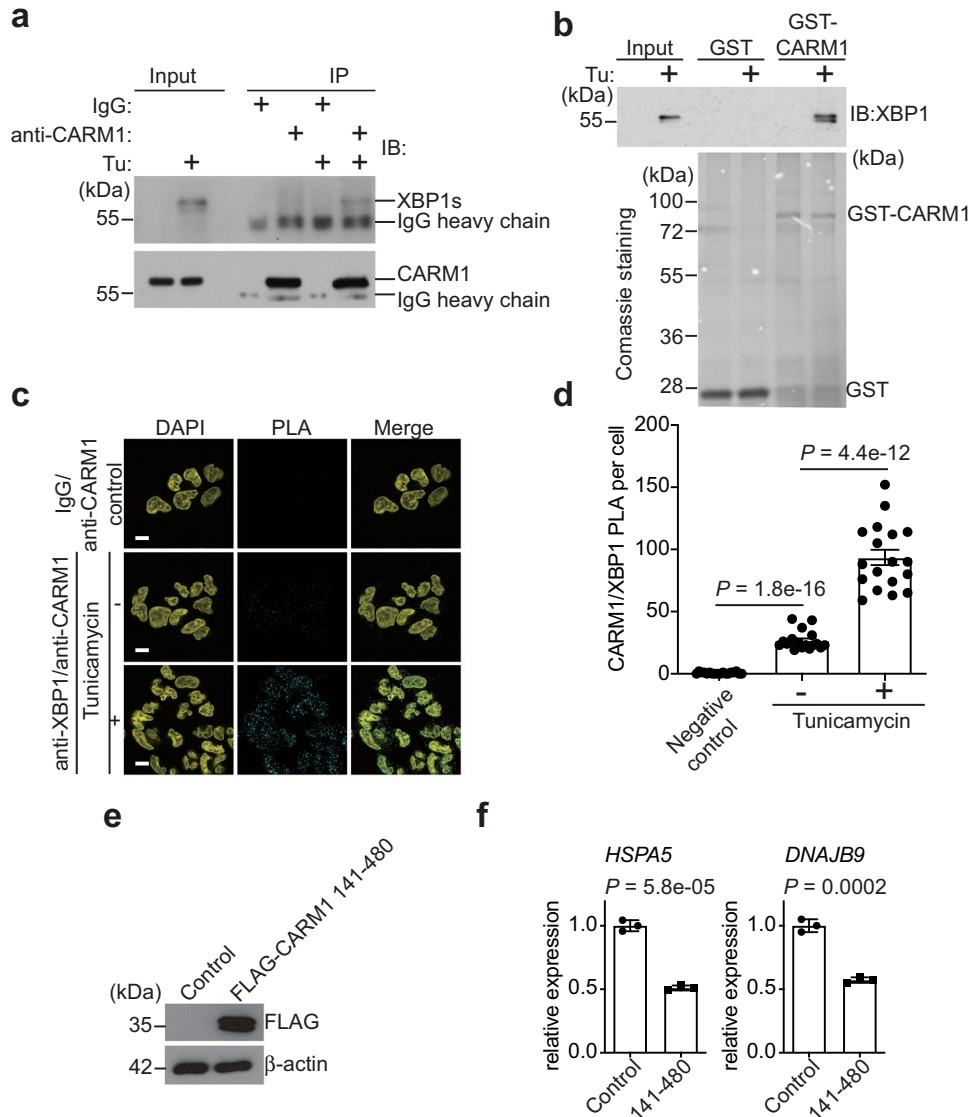

**Fig. 4 CARM1 and XBP1s interacts with each other. a** Co-immunoprecipitation assay using an anti-CARM1 antibody to detect the interaction between CARM1 and XBP1s in A1847 cells treated with or without tunicamycin (5 µg/ml, 8 h). Immunoblots are representative of three biologically independent experiments with similar results. **b** GST-pulldown assay using purified GST-CARM1 and negative control GST in lysates of A1847 cells treated with vehicle control or tunicamycin (5 µg/ml, 8 h). Immunoblots are representative of two biologically independent experiments with similar results. **c** In situ interaction between CARM1 and XBP1s examined by proximity ligation assay (PLA). Isotype-matched IgG and an anti-CARM1 primary antibody were used as negative controls. Scale bar = 10 µm. **d** Quantification of PLA signal in the indicated cells. n = 18. **e** Expression of FLAG-tagged CARM1 aa 141–480 truncation mutant and a loading control β-actin in A1847 cells determined by immunoblot. **f** RT-qPCR analysis of the expression of *HSPA5* and *DNAJB9* in the indicated cells. Data represent mean ± SEM, n = 3 biologically independent experiments. P values were calculated using two-tailed t test.

reduced the tumor burden in the orthotopic A1847 xenograft model (Fig. 6a, b). This correlated with a significant improvement of survival of mice bearing CARM1-expressing A1847 tumors (Fig. 6c). As a control, B-I09 did not significantly reduce the tumor burden in the orthotopic model established by CARM1 knockout A1847 cells (Supplementary Fig. 5b). Likewise, B-I09 significantly reduced the tumor burden in CARM1-high, but not in CARM1-low, HGSOC patient-derived xenograft (PDX) models (Fig. 6d, e and Supplementary Fig. 5c, d). Notably, B-I09 was well tolerated. For example, B-I09 treatment did not significantly decrease the body weight of the treated mice, indicating that the treatment was not toxic (Supplementary Fig. 5e).

We next sought to correlate the observed tumor-suppressive effects in vivo with the mechanism we have characterized in vitro. Toward this goal, we examined the expression of a cell proliferation marker Ki67 and an apoptosis marker cleaved caspase 3. Indeed, B-I09 treatment significantly decreased Ki67 expression, while it increased cleaved caspase 3 expression in CARM1-high tumors (Fig. 6g, h). Together, these data support that IRE1α inhibitor B-I09 suppresses CARM1-high HGSOC in vivo, which correlates with the induction of apoptosis and suppression of cell proliferation.

**IRE1α inhibition synergizes with immune checkpoint blockade.** In addition to promoting survival of cancer cells, ER stress response such as the IRE1α/XBP1s pathway plays a critical role in eliciting an immune-suppressive tumor microenvironment[11]. This raises the possibility that inhibition of the IRE1α/XBP1s pathway may synergize with immune checkpoint blockade

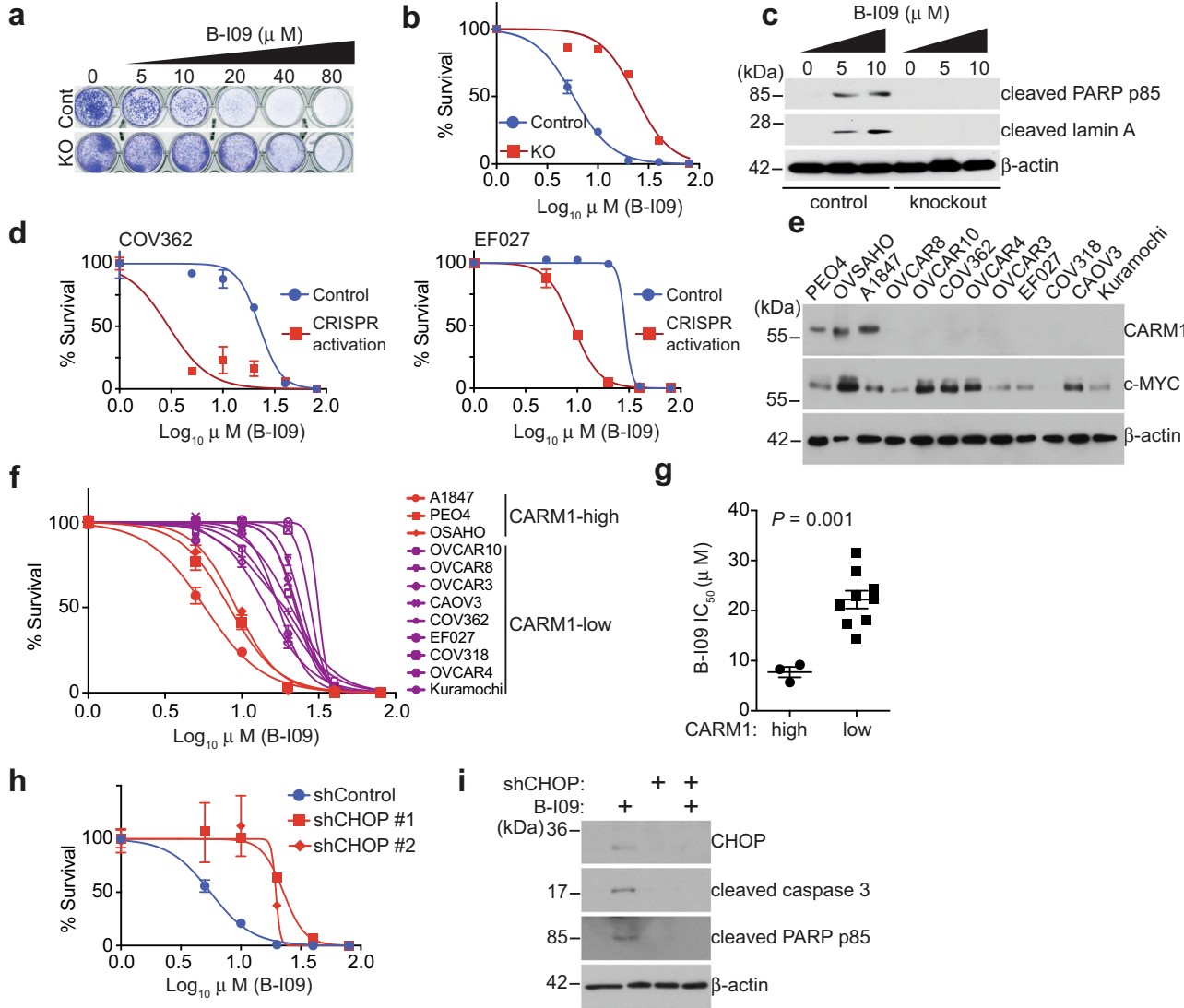

**Fig. 5 CARM1 status correlates with response to inhibition of the IRE1α/XBP1s pathway. a** Representative images of colonies formed by control and CARM1 knockout A1847 cells treated with or without IRE1α inhibitor B-I09 at the indicated concentrations. **b** B-I09 dose-responsive curves based on the colony-formation assay for control and CARM1 knockout A1847 cells. **c** Expression of apoptotic markers cleaved PARP and cleaved Lamin A and a loading control β-actin in control and CARM1 knockout A1847 cells treated with or without B-I09 at the indicated concentrations determined by immunoblot. Immunoblots are representative of three biologically independent experiments with similar results. **d** B-I09 dose-responsive curves based on colony-formation assay of the indicated control and CRISPR-mediated activation of endogenous CARM1-expressing COV362 and EF027 cells. **e** Expression of CARM1, c-MYC, and a loading control β-actin in a panel of the indicated cell lines determined by immunoblot. Immunoblots are representative of two biologically independent experiments with similar results. **f** B-I09 dose-responsive curves based on colony-formation assay for the indicated cells. **g** B-I09 IC$_{50}$ of CARM1-high ($n = 3$) and CARM1-low ($n = 9$) cell lines. **h** B-I09 dose-responsive curves based on colony-formation assay of the indicated control and shCHOP-expressing A1847 cells. **i** Expression of apoptotic markers cleaved caspase 3, cleaved PARP p85, and a loading control β-actin in control and shCHOP-expressing (#1) A1847 cells treated with or without B-I09 (5 μg/ml, 48 h). Immunoblots are representative of two biologically independent experiments with similar results. Data represent mean ± SEM, $n = 4$ biologically independent experiments unless otherwise stated. $P$ values were calculated using two-tailed $t$ test.

therapy[10]. To test whether IRE1α inhibition is synergistic with immune checkpoint blockade, we utilized a syngeneic immune-competent mouse model using UPK10 mouse ovarian cancer cell line[23] that expresses CARM1 at a comparable level as those observed in A1847 cells (Fig. 7a). The orthotopically transplanted tumors were randomized into the following treatment groups: Vehicle/IgG, B-I09/IgG, Vehicle/anti-PD1, and B-I09/anti-PD1 (Fig. 7b). Consistent with our findings from orthotopic xenograft and PDX models, B-I09 treatment significantly reduced the tumor burden in the syngeneic model as a single agent (Fig. 7c, d). Notably, a combination of B-I09 and anti-PD1 was significantly more effective in reducing the tumor

burden compared with either one of the individual treatments (Fig. 7c, d). Consistently, the combination significantly improved the survival of tumor-bearing mice compared with each of the individual treatments (Fig. 7e). As a control, B-I09 treatment did not further reduce the burden of tumors formed by CARM1 knockout UPK10 cells treated with anti-PD1 (Supplementary Fig. 6a, b). B-I09 treatment significantly decreased Xbp1s levels in both tumors and sorted T cells, indicating that B-I09 affects both tumors and the immune cells in the tumor microenvironment (Supplementary Fig. 6c–f). Indeed, immune cell profiling revealed that B-I09 significantly increased the intra-tumoral infiltration of B cells and CD4 T cells

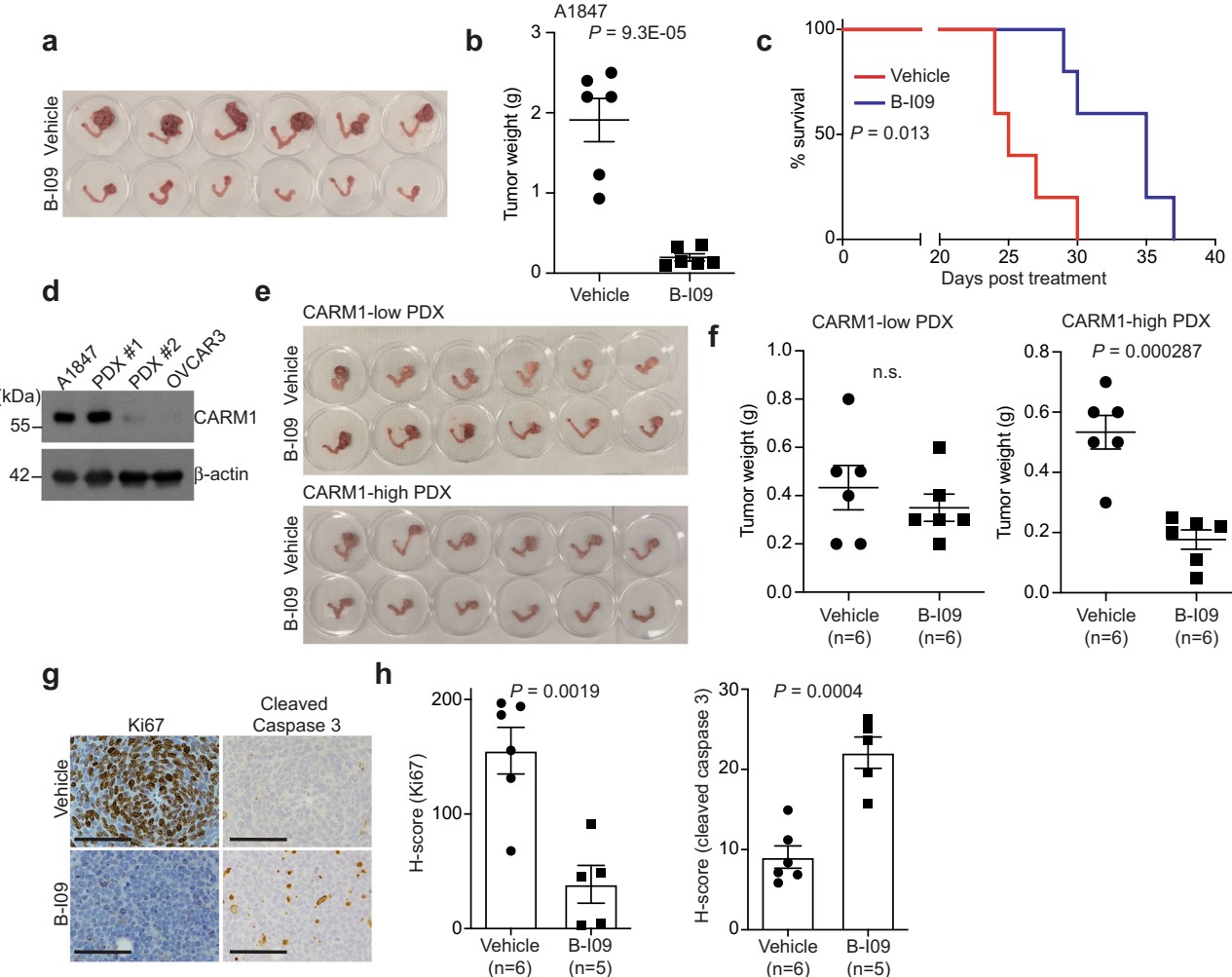

**Fig. 6 CARM1 status correlates with response to IRE1α inhibitor B-I09 in vivo. a, b** A1847 cells were unilaterally injected into the ovarian bursa sac of immunocompromised mice. Tumor-bearing mice were randomized and treated with vehicle control or B-I09 (50 mg/kg, 5 weekdays/week by i.p.) for 2 weeks. After treatment, reproductive tracts with tumors from the indicated treatment groups were dissected (**a**). And tumor weights were measured as a surrogate for tumor burden (**b**). $n = 6$ mice per group. **c** After stopping the treatment, the mice were followed for survival. Kaplan–Meier survival curves for the indicated groups ($n = 5$ mice per group). $P$ value was calculated by log-rank test. **d** Expression of CARM1 and a loading control β-actin in the indicated CARM1-high and CARM1-low PDXs determined by immunoblot. Immunoblots are representative of two biologically independent experiments with similar results. A1847 and OVCAR3 cells were used as controls for CARM1-high and CARM1-low expression, respectively. **e, f** Mice bearing the indicated orthotopic CARM1-high and CARM1-low HGSOC PDXs were randomized and treated with vehicle control or B-I09 (50 mg/kg, 5 weekdays/week by i.p.) for 2 weeks. Reproductive tracts with tumors from the indicated treatment groups were dissected (**e**). And tumor weights were measured (**f**). **g** Tumors from CARM1-high group were subjected to immunological staining for an apoptotic marker cleaved caspase 3 and a cell proliferation marker Ki67. Scale bar = 100 μm. **h** Histological score ($H$ score) of the indicated markers was quantified from three separate fields of each tumor, six tumors from vehicle group, and five tumors from B-I09 group were analyzed. Data represent mean ± SEM. $P$ values were calculated using two-tailed $t$ test except for **c** by log-rank test.

(Fig. 7f and Supplementary Fig. 6g). Although CD8 T cell infiltration was not affected by B-I09 treatment, activation of both CD4 and CD8 T cells was increased by B-I09 treatment (Fig. 7f). Consistent with previous reports[19,24], B-I09 treatment decreased both intra-tumoral and spleen monocytic myeloid-derived suppressor cell (MDSC) and polymorphonuclear MDSC (Fig. 7f and Supplementary Fig. 6h). The combination treatment did not significantly decrease the body weight of the treated mice, and histological analyses of the liver, kidney, and spleen harvested at the end of the experiment from combination-treated mice revealed no overt abnormalities (Supplementary Fig. 6i, j), suggesting that the combination treatment was well tolerated. Together, we conclude that inhibition of the IRE1α/XBP1s pathway is synergistic with immune checkpoint blockade in CARM1-expressing EOCs in vivo.

## Discussion

Hyperactivation of the ER stress response in cancer cells is a therapeutic vulnerability that is actively explored[25]. However, the mechanisms underlying hyperactivation of the ER stress response are not fully understood. We show that CARM1 functions a co-activator of XBP1s in determining the expression of the IRE1α/XBP1s pathway target genes. Thus, our data support a model whereby CARM1 determines the ER stress response by controlling XBP1s's association with its target genes. Notably, it has been reported that XBP1s controls the hypoxia-inducible factor-1α (HIF1α) transcriptional program in triple-negative breast cancer where XBP1s functions upstream of HIF1α[26]. Our results identify CARM1 as a critically important co-activator that functions upstream of XBP1s to drive the IRE1α/XBP1s pathway.

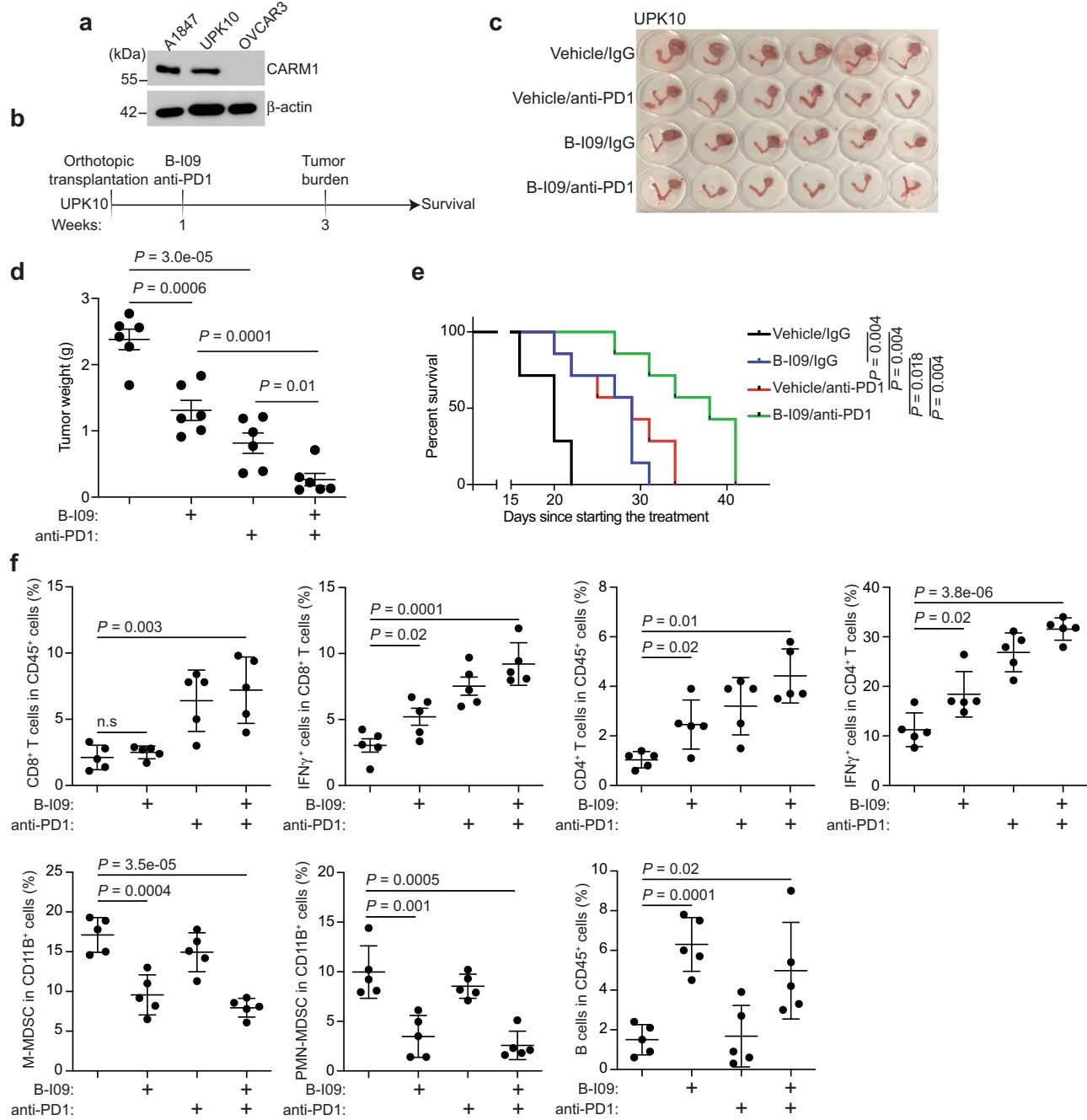

**Fig. 7 B-I09 synergizes with immune checkpoint blockade. a** Expression of CARM1 and a loading control β-actin in UPK10, A1847, and OVCAR3 cells determined by immunoblot. Immunoblots are representative of three biologically independent experiments with similar results. **b** Schematic of the experimental design for UPK10 syngeneic immunocompetent mouse model. **c** Mice bearing orthotopic tumors were randomized into four indicated treatment groups. Reproductive tracts with tumors from the indicated treatment groups were dissected at the end of treatment ($n = 6$ mice per group). **d** The weights of tumors dissected from the indicated groups were measured as a surrogate for tumor burden. Data represent mean ± SEM. $P$ values were calculated using two-tailed $t$ test. Statistical co-efficiency of drug interaction (CDI) analysis revealed that the CDI for the combination is 0.59 (<1)[38], which indicates synergy between B-I09 and anti-PD1 combination. **e** After stopping the treatment, the mice were followed for survival. Kaplan–Meier survival curves for the indicated groups ($n = 7$ per group). **f** Immune cell infiltration in the tumors dissected from the indicated treatment groups were analyzed by flow cytometry. $n = 5$ mice per group. $P$ values were calculated by two-sided log-rank test.

CARM1 is a well-studied transcriptional co-activator through its enzymatic substrates, most notably epigenetic regulators[3,4]. We show that CARM1 interacts with XBP1s through its catalytic domain. However, the enzymatic activity of CARM1 is not required for the expression of CARM1/XBP1s target genes. Thus, CARM1 may function as a scaffold factor to facilitate XBP1s recruitment/association with its target genes. In addition to functioning as a co-activator of gene transcription, CARM1 indirectly promotes the silencing of the EZH2/polycomb repressive complex 2 (PRC2) target genes by regulating the antagonism between PRC2 and the SWI/SNF chromatin remodeling complex[3,9,27]. Thus, CARM1 promotes cancer by both directly enhancing the activation of the IRE1α/XBP1s pathway and indirectly mediating the silencing of EZH2 target genes. One

limitation of our studies is that, in addition to the IRE1α/XBP1s pathway, CARM1 may contribute to the activation of additional pathways including the c-MYC pathway[3]. Regardless, future studies will examine the relative contribution and cooperation among these CARM1-regulated pathways.

In summary, our results show that CARM1-expressing HGSOCs are hypersensitive to inhibition of the IRE1α/XBP1s pathway alone or in combination with immune checkpoint blockade. These findings suggest that CARM1 status could serve as a biomarker to predict response to inhibitors of the IRE1α/XBP1s pathway. CARM1 amplification/overexpression occurs in ~20% of HGSCOs and is mutually exclusive with deficiency in homologous recombination caused by genetic alterations such as BRCA1/2 mutations[9]. Thus, therapeutic approaches for CARM1-expressing HGSOCs are an unmet clinical need because these patients are unlikely to benefit from either standard-of-care platinum-based chemotherapy or emerging PARP inhibitors. Our findings indicate that inhibition of the IRE1α/XBP1s pathway alone or in combination with immune checkpoint blockade represents a therapeutic strategy for a number of cancer types with frequent CARM1 overexpression including HGSOCs[6–8]. Therefore, our findings will have broad implications in developing therapeutic approaches with precision based on CARM1 expression status.

## Methods

### Animals, cell culture, transfection, and reagents.
Six-week-old female immunodeficiency gamma (NSG) mice were obtained from Wistar Institute Animal Facility. C57BL/6 mice were purchased from Charles River Laboratories. Mice were maintained at 22–23 °C with 40–60% humidity and a 12-h light/12-h dark cycle. Human HGSOC cell lines A1847, PEO4, OVSAHO, OVCAR10, OVCAR8, OVCAR4, OVCAR3, CAOV3, COV362, COV318, EF027, Kuramochi, and mouse ovarian cancer cell lines ID8 and UPK10 were cultured in RPMI1640 supplemented with 10% fetal bovine serum (FBS) and 1% penicillin/streptomycin at 37 °C with 5% CO$_2$. Viral packaging HEK293T cells were cultured in Dulbecco's modified Eagle's medium with 10% FBS and 1% penicillin/streptomycin at 37 °C supplied with 5% CO$_2$. All the cell lines were authenticated at The Wistar Institute Genomics Facility using short tandem repeat DNA profiling. Mycoplasma testing was performed using LookOut Mycoplasma PCR detection (Sigma) every month. Transfection was performed using Lipofectamine 2000 (Life Technologies) following the manufacturer's instructions. Small-molecule drugs used in this study are as follows: tunicamycin (Fisher Scientific, #351610), 4μ8c (APExBIO, #B1874), B-I09 (a gift from Professor Chih-Chi Andrew Hu, Houston Methodist Research Institute), and EZM 2302 (ProbeChem, #PC-61030).

### Antibodies.
For western blotting, ChIP, and PLA, the following primary antibodies were used: rabbit anti-CARM1 (Cell Signaling, #3379S, 1:1000 dilution), mouse anti-CARM1 (Cell Signaling, #12495, 1:1000 dilution), rabbit anti-XBP1 (Cell Signaling, #12782S, 1:1000 dilution), anti-cleaved PARP (Cell Signaling, #5625S, 1:1000 dilution), mouse anti-CHOP (Cell Signaling, #2895, 1:1000 dilution), rabbit anti-lamin A/C (Cell Signaling, #2032S, 1:1000 dilution), mouse anti-β-actin (Sigma, #A5316, 1:1000 dilution), rabbit anti-flag-M2 (Cell Signaling, #2368S, 1:1000 dilution), rabbit anti-H3R17me2a (Abcam, #ab8284, 1:1000 dilution), mouse anti-histone H3 (Cell Signaling, #14269S, 1:1000 dilution), and rabbit anti-XBP1 (Novus Biologicals, #NBP1-77681, 1:1000 dilution). For flow cytometry, the following primary antibodies were used: mouse CD45 (Biolegend, #103147, 1:1000 dilution), mouse CD3(BD, #552774, 1:1000 dilution), mouse CD4 (Biolegend, #100516, 1:1000 dilution), mouse CD8 (Biolegend, #100708, 1:1000 dilution), mouse CD11C (Biolegend, #117324, 1:1000 dilution), mouse CD11B (Biolegend, #101259, 1:1000 dilution), mouse B220 (Biolegend, #103227, 1:1000 dilution), mouse CD19 (Biolegend, #115523, 1:1000 dilution), mouse Ly6C (Biolegend, #128026, 1:1000 dilution), and mouse Ly6G (Biolegend, #127639, 1:1000 dilution). For immunohistochemistry (IHC), cleaved caspase 3 (Cell Signaling, #9661, 1:200 dilution) and Ki67 (Cell Signaling, #9449, 1:1000 dilution) were used.

### DNA constructs.
To construct vectors expressing GST-tagged CARM1 and CARM1 truncation mutants, CARM1 and truncation mutants were amplified by PCR using primers as detailed in Supplementary Table 1. Lentiviral EX-Y3476-Lv105 encoding human CARM1 expression construct obtained from Genecopoeia was used as PCR template. The amplified DNA fragments were digested with EcoRI and NotI and subcloned to the same sites of pGEX-6p-1 backbone plasmid.

To construct the pLentiCRISPR-CARM1 plasmid for CARM1 knockout, pLentiCRISPR v2 (Addgene #52961) was digested with BsmBI (NEB) at 55 °C for 1 h and run on a 1% agarose gel. The digested plasmid was cut out and purified using the QIAquick Gel Extraction Kit (Qiagen, 166047244). Each pair of oligos

were phosphorylated using T4 PNK (M0201S) in T4 ligation buffer (New England Biolabs) and annealed in a thermocycler at 37 °C for 30 min, 95 °C for 5 min, ramped down to 25 °C at 5 °C/min. Annealed oligonucleotides were diluted 1:200 in RNase/DNase-free water. Ligation of the annealed oligonucleotide and digested pLentiCRISPR v2 plasmid was performed using Quick Ligase (New England Biolabs). CARM1 guide RNA (5′-AGCACGGAAAATCTACGCGG-3′) was used for cloning[9]. The scramble control shRNA (#1864) and viral packaging plasmids pMD2.G (#12259) and psPAX2 (#12260) were obtained from Addgene. TRC lentiviral vectors (TRCN0000019807) encoding shRNA against human XBP1 was obtained from Molecular Screening Facility at the Wistar Institute.

### Immunoblots.
Cells were trypsinized and washed two times with phosphate-buffered saline (PBS). Protein was extracted with RIPA lysis buffer (50 mM Tris (pH 8.0), 150 mM NaCl, 1% Triton X-100, 0.5% sodium deoxycholate, 0.1% sodium dodecyl sulfate (SDS) and 1 mM phenylmethylsulfonyl fluoride (PMSF)). Protein concentration was measured using the BCA assay (Pierce). Samples were separated by SDS-polyacrylamide gel electrophoresis (SDS-PAGE) and transferred to polyvinylidene fluoride membrane (Millipore). Membranes were blocked with 5% non-fat milk and then incubated with primary and secondary antibodies.

### Reverse-transcriptase qPCR.
Total RNA was extracted using the RNeasy Kit (Qiagen) according to the manufacturer's instructions. Purified RNA was used for RT-PCR with the High-Capacity cDNA Reverse Transcription Kit (Thermo fisher). qPCR was performed using iTaq Universal SYBR Green Supermix (Bio-Rad) and run on QuantStudio 3 Real-Time PCR System. The primers sequences used for qRT-PCR are listed in Supplementary Table 1.

### Lentivirus infection.
pLenti-CRISPR v2 (Addgene #52961)[28] and pLVX systems were used for lentivirus package. HEK293FT cells were transfected by Lipofectamine 2000. Lentivirus was harvested and filtered with 0.45 μm filter at 72 h post transfection. Cells infected with lentiviruses for 48 h were selected in medium contains 1 μg/ml puromycin.

### CRISPR-mediated activation of endogenous CARM1.
Human CARM1 CRISPR lentiviral activation particles were purchased from Santa Cruz (#sc-404087-LAC). Cells were infected with CARM1 CRISPR lentiviral activation particles in complete medium with 5 μg/ml polybrene for 24 h, followed by selection in complete medium containing 2 μg/ml puromycin, 200 μg/ml hygromycin B, and 2 μg/ml Blasticidin S HCl for 1 week.

### Colony-formation assay.
Three thousand cells were plated into a 24-well tissue culture plate and treated with the indicated compounds. Cell medium was changed every 3 days with the indicated drug doses for 10 days. Colonies were visualized by staining the plates with 0.05% crystal violet. Integrated density was determined using the NIH ImageJ software.

### ChIP and ChIP-reChIP.
For ChIP analysis[29], cells were cross-linked with 1% formaldehyde/PBS for 10 min at room temperature and then quenched by 0.125 M glycine for 5 min. Fixed cells were lysed with ChIP lysis buffer 1 [50 mM Hepes-KOH (pH 7.5), 140 mM NaCl, 1 mM EDTA (pH 8.0), 1% Triton X-100, and 0.1% deoxycholate] on ice and lysis buffer 2 [10 mM Tris (pH 8.0), 200 mM NaCl, 1 mM EDTA, and 0.5 mM EGTA] at room temperature. Chromatin was digested with MNase (Cell Signaling Technology) in digestion buffer [10 mM Tris (pH 8.0), 1 mM CaCl$_2$, and 0.2% Triton X-100] at 37 °C for 15 min. The nucleus was broken down by one pulse (30 s) of bioruptor with high output. After centrifugation, the digested chromatin was collected and incubated overnight at 4 °C, and Protein A/G Dynabeads were added to the reaction for another 1.5 h. Magnetic beads were washed, and chromatin was eluted in TES buffer (1% SDS, 1 mM EDTA, 10 mM Tris-Cl pH 8.0) or TE/10 mM dithiothreitol (DTT) for ChIP-reChIP. Eluted DNA/protein was treated with proteinase K at 55 °C for 45 min and decross-linked at 65 °C overnight. A Zymo ChIP DNA Clean and Concentrator Kit (Zymo Research, catalog no. D5205) was used to purify the DNA for qPCR.

For ChIP-reChIP[30], following the step of washing magnetic beads in ChIP as describe above, the proteins were eluted with TE/10 mM DTT. The eluted samples were diluted 20 times with dilution buffer (1% Triton X-100, 2 mM EDTA, 20 mM Tris-HCl pH8.1, 150 mM NaCl) and incubated with the indicated antibodies for second round IP overnight at 4 °C. Protein A/G Dynabeads were added to the reaction for another 1.5 h. Magnetic beads were washed, and chromatin was eluted in TES buffer (1% SDS, 1 mM EDTA, 10 mM Tris-Cl pH 8.0). Eluted DNA/protein complex was treated with proteinase K at 55 °C for 45 min and decross-linked at 65 °C overnight. A Zymo ChIP DNA Clean and Concentrator Kit (Zymo Research, catalog no. D5205) was used to purify the DNA for qPCR.

The following antibodies were used: rabbit anti-CARM1 (Cell Signaling, #3379S) and rabbit anti-XBP1 (Novus Biologicals, #NBP1-77681). An isotype-matched IgG was used as a negative control. ChIP DNA was used for ChIP-qPCR and ChIP-seq. For ChIP-qPCR against the promoters of target genes, the primers are listed in Supplementary Table 1.

**Reporter assay**. A1847 cells ($1 \times 10^5$ per well) were seeded in 12-well plate and co-transfected with 180 ng pGL4-UPRE-luc2P-Hygro (Addgene #101788) and 20 ng pRL-SV40 (Promega, plasmid no. E223A) using Lipofectamine 2000 (Invitrogen) overnight. The next day, cells were treated with or without 5 μg/ml tunicamycin for 24 h and assayed for luminescence using a Dual-Luciferase Reporter Assay System (Promega, #E1910). Luminescence was measured using a Victor X3 2030 Multi-label Reader (Perkin Elmer).

**IP and GST-pulldown assay**. A1847 cells were treated with or without 5 μg/ml tunicamycin for 8 h. Cells were collected and resuspended in buffer A (0.1% Triton X-100, 20 mM Tris-HCl pH7.9, 10 mM KCl, 1.5 mM $MgCl_2$, 0.5 mM PMSF) containing 300 mM NaCl on ice for 10 min and rotated at 4 °C for 30 min followed by centrifuge at $13,000 \times g$, 4 °C for 15 min. The supernatants were recovered and diluted with twice volumes of buffer A without NaCl. The cell extracts were used for IP and GST-pulldown assays.

For IP, the extracts were incubated with 4 μg control IgG or anti-CARM1 (Cell Signaling, #3379S) antibody for 3 h at 4 °C followed by incubation with protein A + G Dynabeads for another 1 h. After incubation, the beads were washed with buffer A containing 150 mM NaCl for three times, and immunoprecipitated proteins were eluted by glycine buffer (0.2 M glycine, pH 2.5), separated by SDS-PAGE, and detected by western blotting.

For GST-pulldown, GST, GST-CARM1, or GST-CARM1 truncation mutants were incubated with glutathione-sepharose beads in buffer A containing 150 mM NaCl for 45 min and washed with the same buffer twice. The beads were mixed with cell extracts and incubated at 4 °C for 4 h followed by extensive washing with the same buffer. Proteins were eluted from the beads by an SDS sample buffer, separated by SDS-PAGE, and detected by western blotting.

**CUT & RUN sequencing**. For CUT&RUN sequencing[31], cells were harvested by trypsinization and gently washed twice using wash buffer (20 mM HEPES pH 7.5, 150 mM NaCl, 0.5 mM spermidine, and EDTA-free Protease Inhibitor Cocktail). Cells were then incubated with the antibody at 4 °C overnight in antibody buffer (wash buffer supplemented with 0.05% digitonin and 2 mM EDTA). The next day, supernatant was removed by centrifugation and cell pellets were washed once with Dig-wash buffer (Wash buffer containing 0.05% digitonin). Cell pellets were then incubated with Protein A MNase (700 ng/ml in Dig-wash buffer) for 1 h by rotation at 4 °C. After three times of washes, cell pellets were resuspended in 100 μl Dig-wash buffer with 2 μl 100 mM $CaCl_2$ and incubated at 0 °C for 30 min; reactions were stopped by addition of 100 μl 2× STOP buffer (340 mM NaCl, 20 mM EDTA pH 8.0, 4 mM EGTA, 0.05% digitonin, 50 μg/ml RNase A, 50 μg/ml glycogen). The supernatant DNA was collected after centrifugation and further purified using phenol–chloroform–isoamyl alcohol (Sigma, #145 p3803) extraction and ethanol precipitation. Purified DNA was used for library construction using the NEBNext Ultra DNA Library Prep Kit (NEB, E7645) following the manufacturer's instructions, and the libraries were sequenced in a 75-base pair single-end run on the Next Seq 500 (Illumina) at Wistar Genomic facility.

**RNA sequencing**. Control, XBP1 knockdown, and CARM1 knockout A1847 cells were treated with 5 μg/ml tunicamycin or vehicle for 8 h. Total RNA was extracted using the RNeasy Mini Kit (Qiagen, 74106) and digested with DNase I (Qiagen, 79254). RNA-Seq libraries were constructed using the 3′mRNA-seq Library Prep Kit (Lexogen) and sequenced with Illumina NextSeq 500 using 75 nt single read at the Wistar Genomics Facility.

**Proximity ligation assay**. Cells on cover slips were fixed with 3% paraformaldehyde in PBS and permeabilized in PBS containing 0.5% Triton X-100. In Situ PLA Kit (Sigma-Aldrich) was then used to detect protein–protein interaction in fixed, permeabilized cells according to the manufacturer's instructions. The primary antibodies used are rabbit anti-XBP1 (Novus Biologicals, #NBP1-77681) and mouse anti-CARM1 (Cell Signaling, #12495), and an isotype-matched IgG is used as a negative control. Stained slides were analyzed using a Leica TCS SP5 II scanning confocal microscope.

**In vivo mouse models**. Animal protocols were approved by the Institutional Animal Care and Use Committee (IACUC) of The Wistar Institute. NSG mice were purchased from Wistar Institute Animal Facility. C57BL/6 mice were purchased from Charles River Laboratories.

For orthotopic xenograft model in NSG mice, $1 \times 10^6$ A1847 control or A1847 CARM1 knockout cells were unilaterally injected into the ovarian bursa sac of 6–8-week female NSG mice ($n = 11$ mice per group). Ten days after injection, mice were treated with vehicle or B-I09 (50 mg/kg, 5 weekdays per week by i.p.) for 2 weeks. After treatment, tumors from 6 mice per group were surgically dissected and tumor weight was measured. The remaining 5 mice were used for survival experiment and monitored until tumor burden reached 10% of the body weight as determined by The Wistar Institute IACUC guideline.

For immunocompetent syngeneic mouse model in C57BL/6 mice, $1 \times 10^6$ mouse ovarian tumor UPK10 cells were unilaterally injected into the ovarian bursa sac of 6-8-week female C57BL/6 mice (n = 13 mice per group). One week after injection, mice were treated with vehicle, B-I09 (50 mg/kg, 5 weekdays/week by

i.p.), anti-PD1 (BioXCell, #BE0273) (10 mg/kg, twice per week), or a combination for 2 weeks. After treatment, tumors from six mice per group were surgically dissected, and tumor weight was measured. The remaining seven mice were used for survival experiment.

For CARM1-high and CARM1-low PDX models, the procurement of human ovarian tumor tissues was approved by the Institutional Review Board of Christina Care Health System. Third passage of previously established PDXs were orthotopically transplanted into the ovarian bursa sac of 6–8-week-old female NSG mice ($n = 6$ mice per group)[27]. Specifically, CARM1-high PDX was established using a metastatic high-grade serous ovarian carcinoma of Mullerian origin. The cystic mass show sheets of malignant cells with many mitotic figures. CARM1-low PDX was established using a metastatic poorly differentiated papillary serous ovarian carcinoma. Pathological examination reveals the presence of extensive poorly differentiated papillary serous carcinoma. PDX-bearing mice were randomized once the tumor was established and treated with vehicle (20–30 μl dimethyl sulfoxide (DMSO)) or B-I09 (50 mg/kg dissolved in 20–30 μl DMSO, 5 weekdays/week by i.p.) for 2 weeks as was previously published[15]. After treatment, tumors were surgically dissected, and tumor weight was measured as a surrogate for tumor burden.

**IHC staining**. Tumors were fixed in phosphate-buffered 10% formalin (Fisher Scientific, #SF100-4), embedded in paraffin, and cut into serial sections. IHC was performed using Dako EnVision+ system following the manufacturer's instructions. Briefly, the sections were dewaxed, rehydrated, and immersed in 3% hydrogen peroxide in methanol to quench endogenous peroxidase activity. Antigen retrieval was performed in sodium citrate buffer (Thermo Fisher, #005000). The sections were incubated with blocking buffer (PBS supplemented with 1% bovine serum albumin) for 1 h, primary antibody against cleaved caspase 3 (Cell Signaling, #9661, 1:200) or Ki67 (Cell Signaling, #9449, 1:1000) at 4 °C overnight, and secondary antibody for 1 h. Counterstaining was performed using Mayer's Hematoxylin (Dako, #3309S). Expression of the stained markers was scored using a histologic score (H score).

**Flow cytometry**. Tumor was minced into small (1–2 mm) pieces and digested with 1 mg/ml collagenase IV (Sigma-Aldrich, #C5138), 0.1 mg/ml hyaluronidase (Sigma-Aldrich, #H6254), and 0.01 mg/ml deoxyribonuclease I (Sigma-Aldrich, #D5025). The cells were sequentially filtered through 45 μm cell strainer. Single-cell suspensions were prepared, and red blood cells were lysed using ACK Lysis Buffer (Thermo Fisher, #A1049201). Tumor-infiltrated lymphocytes were followed by viability staining (Thermo Fisher, #L34957). Before antibody staining, cells were blocked by an anti-Fcγ receptor antibody (BD, #553142, 1:1000 dilution) and then surface staining was performed in fluorescence-activated cell sorter (FACS) buffer (3% FBS in PBS) with fluorochrome-conjugated antibodies against: mouse CD45 (Biolegend, #103147, 1:1000 dilution), mouse CD3 (BD, #552774), mouse CD4 (Biolegend, #100516, 1:1000 dilution), mouse CD8 (Biolegend, #100708, 1:1000 dilution), mouse CD11C (Biolegend, #117324, 1:1000 dilution), mouse CD11B (Biolegend, #101259, 1:1000 dilution), mouse B220 (Biolegend, #103227, 1:1000 dilution), mouse B cell marker CD19, mouse CD19 (Biolegend, #115523, 1:1000 dilution), mouse Ly6C (Biolegend, #128026, 1:1000 dilution), and mouse Ly6G (Biolegend, #127639, 1:1000 dilution).

For interferon-γ (IFNγ) staining, freshly isolated cells in RPMI with L-glutamine (Thermo Fisher, #25030149) (supplemented with 10% FBS, 1× non-essential amino acids, 1× streptomycin and penicillin, 50 nM/ml β-mercaptoethanol) were stimulated with cell activation cocktail with brefeldin A (Biolegend, #423303) overnight, then the cells in the supernatant were collected for the indicated surface staining, and then intracellular IFNγ staining was performed using an anti-mouse IFNγ antibody (Biolegend, #505840). All FACS analyses were performed on a BD LSR II or a Canto II Flow Cytometer, and data were analyzed with the FlowJo software (Tree Star, Inc.).

**Bioinformatics and statistical analysis**. Cut-and-run data were aligned using bowtie[32] against hg19 version of the human genome and HOMER[33] was used to generate bigwig files with default normalization parameters and call significant binding peaks for CARM1 and XBP1s vs. input control using options "-style factor". The default normalization parameter is to convert alignment to values that are the number of tags per 1 bp per 10 M reads. Peaks that passed FDR < 5% with at least fourfold over control threshold were considered significant. Normalized binding signals were derived from bigwig files using bigWigAverageOverBed tool from UCSC toolbox[34] with mean0 option over peak region for peak signal and 60 bp window centered at peak center for heatmaps and line plots. Fold differences between samples were calculated with average input signal 0.4 (average background input value) used as a floor for the minimum allowed signal. Distance from TSS of 1 kb was used for gene-peak assignments.

RNA-seq data was aligned using bowtie2[35] against hg19 version of the human genome and RSEM v1.2.12 software[36] was used to estimate raw read counts and RPKM values using Ensemble transcriptome. DESeq2[37] was used to estimate significance of differential expression between group pairs and calculate normalized counts for heatmaps. Overall gene expression changes were considered significant

if it passed FDR < 5% thresholds unless stated otherwise. Significance of overlap between sets of genes was estimated using hypergeometric distribution test.

For statistical analyses, experiments were repeated at least three times unless otherwise stated. Statistical analysis was performed using the GraphPad Prism 7 (GraphPad) software. Quantitative data are expressed as mean ± SEM unless otherwise stated. For all statistical analyses, the cutoff for significance was set at 0.05. For correlation studies, Pearson's correlation was used for calculating $P$ and $R$ values in Microsoft Excel. Animal experiments were randomized. For drug dose-response survival analysis, all drug doses were $\log_{10}$ transformed before data analysis to improve normality and homoscedasticity. For in vivo mouse experiment, analysis of variance with post hoc multiple comparisons were performed for between-group comparisons. The coefficient of drug interaction (CDI) was used to determine whether the combination of two drugs is synergistic in vivo[38]. The CDI is calculated as follows: $CDI = AB/(A + B)$. $AB$ is the ratio of the combination groups to the control group; and $A$ or $B$ is the ratio of the single agent group to control group.

**Reporting summary**. Further information on research design is available in the Nature Research Reporting Summary linked to this article.

## Data availability

The cut-and-run and RNA-seq data generated in this study have been deposited in the Gene Expression Omnibus (GEO) under accession GSE157118. TCGA HGSOC RNA-seq dataset was downloaded from cBioPortal (https://www.cbioportal.org/). Cancer Cell Line Encyclopedia RNA-seq data were downloaded from (https://sites.broadinstitute.org/ccle/datasets). hg19 was downloaded from UCSC (https://hgdownload.soe.ucsc.edu/downloads.html). Source data are provided with this paper.

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

## Acknowledgements

This work was supported by US National Institutes of Health grants (R01CA160331, R01CA163377, R01CA202919, R01CA239128, and P01AG031862 to R.Z.; P50CA228991 to R.Z.; K99CA241395 to S.K.; R01CA163910 and R01CA190860 to C.-C.A.H.; DP2-NS105575 to K.S.; and F31CA247336 to J.Z.) and US Department of Defense (OC180109 and OC190181 to R.Z.); The Honorable Tina Brozman Foundation for Ovarian Cancer Research to R.Z.; and the Ovarian Cancer Research Alliance (Collaborative Research Development Grant to R.Z. and Ann and Sol Schreiber Mentored Investigator Award to S.W. and J.L.). Support of Core Facilities was provided by Cancer Center Support Grant (CCSG) CA010815 to The Wistar Institute.

## Author contributions

J.L., H.L, T.F., J.Z., Q.Y., S.W., W.Z., D.G. and S.K. performed the experiments and analyzed data. A.V.K. performed the bioinformatic analysis. J.L. and R.Z. designed the experiments. C.-H.A.T. and C.-C.A.H. contributed key experimental materials. C.-C.A.H., K.S. and R.Z. supervised the studies. J.L., S.W., S.K., C.-C.A.H., K.S., A.V.K. and R.Z. wrote the manuscript. R.Z. conceived the study.

## Competing interests

The authors declare no competing interests.

**Additional information**

