## [Peer Review File · Nature Communications]

Targeting the IRE1 α /XBP1s pathway suppresses CARM1-expressing ovarian cancerREVIEWER COMMENTS

Reviewer #1 (Remarks to the Author):

In this study, Lin et al. uncover that CARM1 co-operates with the IRE1-XBP1 arm of the UPR to promote progression of ovarian tumors expressing CARM1. The authors report a novel molecular process whereby CARM1 and the IRE1-activated transcription factor XBP1s interact to regulate downstream events in the ER stress response, as well as in cancer cell survival and tumor progression. Further, the study shows that targeting IRE1 using a small-molecule inhibitor of its RNase domain, B-109, can be used to control the growth of CARM1-overexpressing ovarian tumors and also to improve the effects of immune checkpoint blockade. This is an excellent study full of interesting data and relevant findings. The beginning of the paper exploiting cut-and-run approaches to identify the potential functional interaction between CARM1 and XBP1s is outstanding. A physical interaction between the two factors is then dissected quite elegantly. The computational biology and molecular genetics components of this study are therefore very strong. This is a significant story unearthing a key contribution of CARM1 to the UPR. Hence, the implications of these findings will certainly go beyond cancer biology. The therapeutic aspect of the study is also important, but I have a series of general comments that could be addressed by the authors in order to clarify and/or strengthen some of the claims made in this study:

- 1) Since other reports have demonstrated the functional interaction between c-MYC and IRE1-XBP1 signaling in cancer, could the authors comment/speculate whether CARM1 operates in parallel or upstream of c-MYC to dictate the function of XBP1 in this setting? Along the same lines, what is the status of c-Myc in CARM1-expressing vs. CARM1-non-expressing cells and how could this be affecting IRE1-XBP1 activation in ovarian cancer cells? Indeed, c-MYC was identified by the authors a top regulator in Figure 1C.
- 2) Do CARM1-expressing cancer cells demonstrate a higher basal/constitutive level of IRE1/XBP1 activation and global ER stress responses compared with cancer cells not expressing CARM1, like it has been demonstrated in breast cancer and lymphoma models overexpressing MYC?
- 3) The authors claim that unresolved ER stress can promote apoptosis. Why would targeting IRE1 with B109 induce apoptosis only in CARM1-expressing cells? Do these cells demonstrate proapoptotic overactivation of the PERK-Chop arm (for instance) upon treatment with B109? The functional concepts of terminal vs. pro-survival ER stress responses coordinated by CARM1 upon IRE1 inhibition need to be described more thoroughly. Is there any role for RIDD in this process?
- 4) The authors present evidence showing that treatment with anti-PD1 and B109 induce maximal therapeutic benefit in tumor-bearing mice. However, these effects are not synergistic. The data indicate that the effects are actually additive. The language therefore needs to be corrected. Also, a major question is whether the additive therapeutic effects of B109 plus anti-PD-1 depend on the expression of CARM1.
- 5) The immunophenotyping presented in Extended Figure 6 is not robust. This is a critical part of the paper that needs to be improved in order to better understand the effects of combination treatment. There are major issues with the gating strategies used and it seems that a high proportion of cells died during tissue processing, hence affecting the recovery of key immune cell types. D Gabrilovich and CC Hu have independently shown that IRE1 controls the influx and function of MDSCs in cancer hosts. What happens with MDSCs/neutrophils in the TME or distally upon treatment with B-109 in the model used? Also, what is the activation status or effector profile of TILs upon treatment with B109? While TIL infiltration did not change, could treatment with B-109 plus anti-PD1 modulate the anti-tumor phenotype of these T cell populations to drive adaptive anti-tumor immune control?
- 6) It would be relevant to provide some evidence of target engagement in the TME upon treatment with B109. For instance, is the compound targeting IRE1 predominantly in cancer cells or also immune cells of the TME? If so, how strong and sustained is the inhibition of IRE1-XBP1 signaling (e.g. reduction in Xbp1s levels) in these cells upon daily administration with B109? Along the same lines, the authors should describe the vehicle used to administer B109 into tumor-bearing mice as this information is lacking.

Reviewer #2 (Remarks to the Author):

High grade serous ovarian cancer (OC) is a deadly disease for which new therapeutic approaches are badly needed. Moreover, personalized therapeutic strategies for OC are lacking at this time. Dr. Zhang and colleagues extend their previously described role of CARM1, which they reported to be amplified/overexpressed in ~20% of OC, in both of these critically important areas. They examined CARM1-expressing OC cells in the context of endoplasmic reticulum (ER) stress response and the IRE1 α /XBP1 pathway. The authors demonstrated for the first time that pharmacological targeting of the IRE1 α /XBP1 pathway selectively suppressed CARM1-expressing OC. The observation that inhibiting the IRE1 α /XBP1 pathway synergizes with immune checkpoint blockade in CARM1-expressing cancers is timely and important. The mechanism of CARM1 determination of the ER stress response was through the IRE1 α /XBP1 pathway, forming a complex with XBP1 to regulate its target gene expression.

The authors used OC cell lines in vitro and in vitro, appropriate PDX (low and high CARM1 expression), analyses of TCGA and Broad Institute databases and state-of-the-art next generation genomics approaches and mouse models to support their conclusions that CARM1-expressing OC cells are selectively sensitive to inhibition of the IRE1 α /XBP1 pathway. They provide comprehensive and compelling novel findings that impacts the OC field and continues to move the field forward in terms of a therapeutic strategy for CARM1-expressing cancers- CARM1 promotes OC by both directly enhancing the activation of the IRE1 α /XBP1 pathway and indirectly mediating the silencing of EZH2 target genes. Furthermore, the findings are important in the context of improving immune-oncology approaches in HGSOC, an unmet need in the field.

The following comments are suggested for consideration by the authors.

Major comments

The majority of the data were generated using A1847 cells with some data using PEO4 cells, such as the CARM1 knockout experiments. The authors suggest that the observed effects are not cell line specific but have the authors confirmed the lack of the ER stress response/XBP1 target genes in the CARM1-null cells or the in vivo results on B109 using a CARM1 null HGSOC cell line as a xenograft? Further clarification would be helpful.

What was the rationale for choosing PEO4 cell line over OVSAHO, which based on the western blot appears to express a much higher level of CARM1 compared to PEO4? Including an explanation would be useful.

Fig 5F: CARM1 band for OVSAHO runs below PEO4 & A1847. A comment would be helpful.

Minor comments

In addition to reference 23, in this manuscript providing a more detailed description of the PDX, such as the original patient characteristics from which the PDXs were derived, would be helpful to the reader.

Line 123 "CAMR1" should be CARM1

Reviewer #3 (Remarks to the Author):

In this manuscript, Lin et al. present data suggesting that CARM1-expressing ovarian cancer cells are selectively sensitive to inhibition of the IRE1 α /XBP1s pathway, both in vitro and in vivo. They find that CARM1 interacts with XBP1s and co-regulates gene expression in ovarian cancer cells, some that are distinct. Furthermore, they show that CARM1 sensitizes ovarian cancer cells to an IRE1 α inhibitor (B-109), either alone or with a checkpoint inhibitor. However, molecular

mechanism(s) of how this happens is not clear. This work is an extension of the authors' earlier work published in Nature Comm. showing that EZH2 inhibition is a potential strategy for to target CARM1 expressing ovarian cancer. The manuscript has significant data that support the role of CARM1 on ER stress signaling and in ovarian cancer. However, there are some significant issues and mechanistic details that are missing.

1. Throughout the text and figures, XBP1 designation is erroneously used for XBP1s; this has to be corrected.
2. In Figure 3, a larger panel of XBP1s-CARM1 target genes should be presented in more than one cell line, to show that what is observed in global analyses can be validated individually.
3. Does a gene expression signature of XBP1s-CARM1 regulated genes have utility as a prognostic marker for ovarian cancer?
4. The data presented on the physical interactions between CARM1 and XBP1s are weak (Figure 4a, Extended Data Figure 3). In Extended Data Figure 3A, IP-Western analysis is shown with the endogenous proteins in A1347 cells, the most robust way to interrogate potential interactions between two factors. The quality of the data in this figure is very low; one can barely discern an XBP1s band among the large background in the specific IP lane. Similarly, in the GST-pulldown assay presented in Extended Data Figure 3C, there are extremely weak bands, despite the large amounts of GST fusion proteins used. An unrelated GST fusion protein can be used in these experiments as a control.
5. In Figure 4A, there are two bands for XBP1s in the pull-down sample, whereas in the other figures (e.g. Figure 2B) a single band is observed. The authors should provide an explanation for this.
6. A previous study from the same group (Karakashev et al., 2018) has shown that CARM1 knockout ovarian cells exhibit significant decrease in growth. Thus, the IC50 values presented in Figure 5 will not properly reflect the sensitivity of a cell line to a certain drug (Hafner et al., 2016, PMID: 27135972). Alternative methods should be employed.
7. In the same study (Karakashev et al., 2018) the authors have shown that CARM1 expressing cells are selectively sensitive to EZH2 inhibition. Is there synergy between EZH2 and IRE1 α inhibition in ovarian cancer cells?
8. The authors have generated CRISPRi-mediated CARM1 activated cells and tested their response to B-I09 (Figure 5). Did they measure IRE1/XBP1s pathway activity in these cells? For example, did CARM1 activation enhance IRE1 α phosphorylation/activation, XBP1 splicing, and XBP1s transcriptional activity including its target gene expression? This information is necessary to properly interpret the data presented.
9. The data provided suggest that CARM1 expressing cells may have enhanced XBP1s activation. Thus, CARM1 expression should correlate with XBP1s activity in patient material; is this so? This analysis can easily be done by scoring for expression of the XBP1s target gene signature and CARM1 expression. The authors should examine whether this correlation is present in multiple publicly available ovarian cancer gene expression datasets.
10. Extended Data Fig. 3e-f: Although inhibition of CARM1 enzymatic activity does not affect the expression of two select CARM1/XBP1 target genes, it cannot exclude the possibility that CARM1 inhibition may affect sensitivity of IRE1 to B-I09.
11. The authors suggest a model whereby CARM1 determines ER stress response by controlling XBP1s association with its target genes. However, direct evidence of this was not provided. Re-ChIP experiments should be provided to interrogate this possibility.
12. Does CARM1-dependent sensitivity to B-I09 rely on any of the CARM1/XBP1s targets identified? Are there other CARM1 downstream effectors (independent of the IRE1-XBP1s pathway) that are implicated in regulating ovarian cancer? A global gene expression experiment and its validation can answer these questions.

Reviewer #4 (Remarks to the Author):

Lin et al in this article performed CUT&RUN experiment to assay distribution of CARM1 and compared this to XBP1 binding and H3K27ac in A1847 cells with and without tunicamycin, an ER stress inducer.

I have several concerns related to the computational analyses of this paper (not in particular order of importance):

- Normalization is important to allow comparison between CUT&RUN experiments, and for

conclusion such as in Fig 1g. How are authors normalizing the data? The authors say "default normalization parameters and call significant binding peaks for CARM1, XBP1 vs input control using options "-style factor". It is not clear what these normalization parameters are, and what approach is used.

- the overlap between DE genes from RNAseq and the CUT&RUN peak mapped genes is not very convincing (Fig 2c), mostly because of the way the analysis is currently done. In Fig 2c, there are 543 DE genes, 10956 XBP1 mapped genes, and 8340 CARM1 mapped genes. Given the number of XBP1 genes are so high (50% of genome), it has a close to 50% random chance to overlap with any gene set of interest. So it renders the overlap of 363 / 543 to be quite meaningless. A better way to analyze is to select equal number of genes from all three groups (i.e. select 500 top XBP1 peak mapped genes by intensity, same for CARM1 top 500 peak-mapped genes), and calculate 3-way overlap. Alternatively, you can lower threshold of DESeq2 to select more DE genes from RNAseq side, but keep each group same size.

- Related to Fig 2c, can authors sort CUT&RUN peak-mapped genes into bins by intensity (or Fold change over vehicle), and plot the number of DE genes that overlap with CUT&RUN genes in each bin? This provides a better assessment of the correlation between CUT&RUN and the targeted gene expression.

- The statistical significance reported in some of the scatterplot heatmaps is at odds with the strength of the correlation (Fig 2e, f). Because you have large sample size, it is very easy to obtain statistical significance. In so doing, one maybe tricked into believing the results are very strong and meaningful, when in fact the correlation value is quite weak (Fig 2f, $R=0.225$, but with $p=7 \times 10^{-5}$). I suggest removing P-values in these plots, and just showing R and perhaps indicate N (sample size). What is the pearson correlation for these scatterplots? Fig 2e does not show a scatterplot that agrees with a $R=0.402$ - the points seem randomly distributed. For both Fig 2e, 2f, can authors switch to density plot (similar to Fig 1f)?

- Have the authors performed Chip-seq of XBP1 and of CARM1 and compared with CUT&RUN to validate their CUT&RUN experiments? What is the advantage of CUT&RUN over Chip-seq for these two factors?

- The authors reported 22,398 CARM1 CUT&RUN peaks, and ?? XBP1 CUT&RUN peaks. I believe that care should be taken in general when interpreting peaks from CUT&RUN experiments, because of the effect of indirect binding (Skene et al, 2017, PMID: 28079019). Direct binding peaks are usually distinguished by the presence of a consensus motif and protection of motif-bound region from pA-MNase enzyme cut due to TF occupancy. Fortunately, direct binding peaks can be teased apart computationally by checking the frequency of cuts within the motif core region and compare with flanking region. This is known as motif footprinting analysis. Can authors perform motif footprinting analysis (Zhu et al, 2019 PMID: 31500663) (Neph et al, 2012, PMID: 22955618) (Pique-Regi et al, 2011, PMID: 21106904), to further confirm that the CUT&RUN peaks are direct binding, and comment the extent of indirect binding? This is critical piece of information for establishing that CARM1 and XBP1 bind to target genes promoters. Since XBP1 has a known motif, it should be expected that targeted gene promoters with XBP1 motif should have no pA-MNase enzyme cuts. So for XBP1 and CARM1: check for the presence of footprints on XBP1 motif.

- RNAseq analyses missing very important volcano plot. It is not clear if the number of DE genes are derived using a combination of significance threshold and fold change values, or just significance value.

- Methods on DESeq2 section of RNASeq analysis: "Overall gene expression changes were considered significant if passed $FDR < 5\%$ thresholds unless stated otherwise." Authors may consider lowering FDR threshold to 10%. 5% maybe considered too stringent.

- Fig 1a, e are missing color scale bar.

- Fig 2d color bars: can authors change the blue end of "fold increase" color scale to use a different color? The color currently overlaps with the $\log(1 + \text{reads}/10M)$ color bar, which is also blue.

Point by Point Response to Reviewers' Comments

We sincerely thank the Reviewers for the constructive and thoughtful review provided for our manuscript. We very much appreciate the Editorial team's clear instructions for revision. All the comments raised are truly valuable to improve the manuscript. Correspondingly, we have strived to fully address their comments. I hope that there is no doubt that we have taken the Reviewers' and Editors' comments very seriously. We believe that by addressing the reviewers' concerns we have produced a more solid and cohesive manuscript. A point-by-point response to the reviewers' comments is detailed below with original comments italicized. Changes that directly address the reviewers' concerns were denoted with vertical lines in the right margin in the revised manuscript. We hope the Reviewers and the Editors will find this manuscript to be much improved and suitable for publication.

Reviewer 1 comments and point-by-point responses: pages 2 – 4

Reviewer 2 comments and point-by-point responses: pages 5 - 6

Reviewer 3 comments and point-by-point responses: pages 7 – 9

Reviewer 4 comments and point-by-point responses: pages 10 - 12

Reviewer #1 (Remarks to the Author):

In this study, Lin et al. uncover that CARM1 co-operates with the IRE1-XBP1 arm of the UPR to promote progression of ovarian tumors expressing CARM1. The authors report a novel molecular process whereby CARM1 and the IRE1-activated transcription factor XBP1s interact to regulate downstream events in the ER stress response, as well as in cancer cell survival and tumor progression. Further, the study shows that targeting IRE1 using a small-molecule inhibitor of its RNase domain, B-109, can be used to control the growth of CARM1-overexpressing ovarian tumors and also to improve the effects of immune checkpoint blockade. This is an excellent study full of interesting data and relevant findings. The beginning of the paper exploiting cut-and-run approaches to identify the potential functional interaction between CARM1 and XBP1s is outstanding. A physical interaction between the two factors is then dissected quite elegantly. The computational biology and molecular genetics components of this study are therefore very strong. This is a significant story unearthing a key contribution of CARM1 to the UPR. Hence, the implications of these findings will certainly go beyond cancer biology. The therapeutic aspect of the study is also important, but I have a series of general comments that could be addressed by the authors in order to clarify and/or strengthen some of the claims made in this study:

1) Since other reports have demonstrated the functional interaction between c-MYC and IRE1-XBP1 signaling in cancer, could the authors comment/speculate whether CARM1 operates in parallel or upstream of c-MYC to dictate the function of XBP1 in this setting? Along the same lines, what is the status of c-Myc in CARM1-expressing vs. CARM1-non-expressing cells and how could this be affecting IRE1-XBP1 activation in ovarian cancer cells? Indeed, c-MYC was

identified by the authors a top regulator in Figure 1C.

Fig.1 for reviewer: MYC is not among the top regulators enriched by CARM1/XBP1s direct target genes. Regulator enrichment analysis of CARM1/XBP1s direct target genes using Ingenuity Pathway Analysis (IPA).

Response: We thank the reviewer for the insightful comments. Our results show that CARM1 enhances the binding of XBP1s' to its target genes. As requested, we now examined the expression of c-MYC in CARM1-expressing and knockout cells and showed that CARM1 knockout did not affect c-MYC expression (**Supplementary Fig. 4g**). We further examined c-MYC expression in the panel of cell lines and our results showed that there is no correlation between CARM1 and MYC expression (**Fig. 5e**). Further, although c-MYC was among the top regulators enriched by CARM1 target

genes, MYC is not among the top regulators enriched by CARM1/XBP1s direct target genes (Fig. 1 for reviewer). This is consistent with previous reports that CARM1 regulates breast cancer migration and metastasis via c-MYC target genes. Thus, our findings support a model whereby c-MYC and XBP1 function in parallel downstream of CARM1.

2) Do CARM1-expressing cancer cells demonstrate a higher basal/constitutive level of IRE1/XBP1 activation and global ER stress responses compared with cancer cells not expressing CARM1, like it has been demonstrated in breast cancer and lymphoma models overexpressing MYC?

Response: We thank the reviewer for the comments. Indeed, our results show that compared with cells not expressing CARM1, CARM1 expression correlates with a higher basal/constitutive level of IRE1/XBP1s pathway based on both XBP1s reporter assay and expression of XBP1s target genes (**Fig. 2g-l and Supplementary Fig. 2d-i**). In addition, XBP1 target genes were significantly downregulated by CARM1 knockout (1462 out of 2990, $P < 10^{-10}$) supporting the notion that CARM1-expressing cancer cells demonstrate a higher basal/constitutive level of IRE1/XBP1 activation (**Supplementary Fig. 2a**).

3) *The authors claim that unresolved ER stress can promote apoptosis. Why would targeting IRE1 with BI09 induce apoptosis only in CARM1-expressing cells? Do these cells demonstrate proapoptotic overactivation of the PERK-Chop arm (for instance) upon treatment with BI09? The functional concepts of terminal vs. pro-survival ER stress responses coordinated by CARM1 upon IRE1 inhibition need to be described more thoroughly. Is there any role for RIDD in this process?*

Fig.2 for reviewer: CARM1 expression does not significantly affect RIDD target genes. $P = 0.2295$. Volcano plot of differential gene expression of RIDD-related genes between control and CARM1 knockout A1847 cells.

Response: We are grateful for the reviewer's insightful comments. As requested, we now show that BI-09 treatment significantly induced CHOP expression at both mRNA and protein levels (**Supplementary Fig. 4k-l**). In addition, we show that knockdown of CHOP expression significantly reduced the IC₅₀ of BI-09 in CARM1-expressing cells, which correlates with a suppression of apoptosis induced by B-I09 (**Fig. 5h-l and Supplementary Fig. 4m**). Thus, as the reviewer rightly predicated, CARM1 coordinates terminal vs. pro-survival ER stress response. In addition, we examined the expression of RIDD target genes in our RNA-seq analysis and we did not observe significant difference in RIDD gene expression between CARM1 expressing and matched isogenic cells (Figure 2 for reviewer). This suggests that RIDD is not implicated in this process.

4) *The authors present evidence showing that treatment with anti-PD1 and BI09 induce maximal therapeutic benefit in tumor-bearing mice. However, these effects are not synergistic. The data indicate that the effects are actually additive. The language therefore needs to be corrected. Also, a major question is whether the additive therapeutic effects of BI09 plus anti-PD-1 depend on the expression of CARM1.*

Response: As requested, to determine whether treatment with the anti-PD1 and BI09 combination is synergistic, we now performed the statistical co-efficiency of drug interaction (CDI) analysis, where CDI value <1 , $= 1$ or >1 indicates synergistic, additive or antagonistic. The CDI for B-I09 and anti-PD1 combination is 0.59, which supports that the combination is synergistic. We now include the information in the legend of Figure 7.

To address the question whether the synergistic effects of B-I09 plus anti-PD1 depends on the expression of CARM1, we now performed new *in vivo* experiments using matched isogenic CARM1 knockout UPK10 cells. Indeed, our new results show that B-I09 did not significantly

reduce the burden of tumors formed by CARM1 knockout UPK10 cells (**Supplementary Fig. 6a-b**). Importantly, the B-I09 and anti-PD1 combination failed to significantly further reduce tumor burden compared with anti-PD1 alone (**Supplementary Fig. 6b**). Thus, the synergistic effects observed in the B-I09 and anti-PD1 combination depends on the expression of CARM1.

5) The immunophenotyping presented in Extended Figure 6 is not robust. This is a critical part of the paper that needs to be improved in order to better understand the effects of combination treatment. There are major issues with the gating strategies used and it seems that a high proportion of cells died during tissue processing, hence affecting the recovery of key immune cell types. D Gabrilovich and CC Hu have independently shown that IRE1 controls the influx and function of MDSCs in cancer hosts. What happens with MDSCs/neutrophils in the TME or distally upon treatment with B-I09 in the model used? Also, what is the activation status or effector profile of TILs upon treatment with BI09? While TIL infiltration did not change, could treatment with B-I09 plus anti-PD1 modulate the anti-tumor phenotype of these T cell populations to drive adaptive anti-tumor immune control?

Response: These are all good points. To address the reviewer's comments, we repeated these experiments once again to improve the gating strategies. Indeed, consistent with previous reports by us and others ^{2,3} and as predicated by the reviewer, B-I09 treatment significantly reduced the infiltrated M-MDSC and PMN-MDSC (**Fig. 7f**). In addition, M-MDSC and PMN-MDSC in distal spleen were also decreased by B-I09 (**Supplementary Fig. 6h**). Further, although infiltrated CD8⁺ T cells were not changed, its activation was increased by B-I09 treatment (**Fig. 7f**). Finally, both infiltration and activation of CD4⁺ T cells were increased by B-I09 treatment (**Fig. 7f**). Thus, the data support that B-I09 plus anti-PD1 modulate the anti-tumor phenotypes of these T cells to drive adaptive anti-tumor immune response.

6) It would be relevant to provide some evidence of target engagement in the TME upon treatment with BI09. For instance, is the compound targeting IRE1 predominantly in cancer cells or also immune cells of the TME? If so, how strong and sustained is the inhibition of IRE1-XBP1 signaling (e.g. reduction in Xbp1s levels) in these cells upon daily administration with BI09? Along the same lines, the authors should describe the vehicle used to administer BI09 into tumor-bearing mice as this information is lacking.

Response: To validate the target engagement in both cancer cells and immune cells, we examined Xbp1s expression in tumors and sorted T cells. Our results show that B-I09 treatment significantly reduced Xbp1 levels in both tumors and sorted T cells (**Supplementary Fig. 6c-f**). As we and others have previously published ⁴, the vehicle control used to administer B-I09 is DMSO. We now include this information in methods section of the manuscript.

Reviewer #2 (Remarks to the Author):

High grade serous ovarian cancer (OC) is a deadly disease for which new therapeutic approaches are badly needed. Moreover, personalized therapeutic strategies for OC are lacking at this time. Dr. Zhang and colleagues extend their previously described role of CARM1, which they reported to be amplified/overexpressed in ~20% of OC, in both of these critically important areas. They examined CARM1-expressing OC cells in the context of endoplasmic reticulum (ER) stress response and the IRE1 α /XBP1 pathway. The authors demonstrated for the first time that pharmacological targeting of the IRE1 α /XBP1 pathway selectively suppressed CARM1-expressing OC. The observation that inhibiting the IRE1 α /XBP1 pathway synergizes with immune checkpoint blockade in CARM1-expressing cancers is timely and important. The mechanism of CARM1 determination of the ER stress response was through the IRE1 α /XBP1 pathway, forming a complex with XBP1 to regulate its target gene expression.

The authors used OC cell lines in vitro and in vitro, appropriate PDX (low and high CARM1 expression), analyses of TCGA and Broad Institute databases and state-of-the-art next generation genomics approaches and mouse models to support their conclusions that CARM1-expressing OC cells are selectively sensitive to inhibition of the IRE1 α /XBP1 pathway. They provide comprehensive and compelling novel findings that impacts the OC field and continues to move the field forward in terms of a therapeutic strategy for CARM1-expressing cancers- CARM1 promotes OC by both directly enhancing the activation of the IRE1 α /XBP1 pathway and indirectly mediating the silencing of EZH2 target genes. Furthermore, the findings are important in the context of improving immune-oncology approaches in HGSOC, an unmet need in the field.

The following comments are suggested for consideration by the authors.

Major comments

The majority of the data were generated using A1847 cells with some data using PEO4 cells, such as the CARM1 knockout experiments. The authors suggest that the observed effects are not cell line specific but have the authors confirmed the lack of the ER stress response/XBP1 target genes in the CARM1-null cells or the in vivo results on B109 using a CARM1 null HGSOC cell line as a xenograft? Further clarification would be helpful.

Response: We thank the reviewer for the comments. Consistent with the notion that the observed changes in CARM1-regulated XBP1s target gene expression is not cell line specific, we validated XBP1s target genes in both A1847 and PEO4 cells with matched isogenic CARM1 knockout cells (**Fig. 2g-h and Supplementary 2d-i**). In addition, we now added new data on isogenic OVSAHO cell lines and show that CARM1 knockout significantly reduced the expression of XBP1s target genes in third cell line (**Supplementary Fig. 2f-g**). In addition, we show that there is a positive correlation between CARM1 and CARM1/XBP1 target genes in the Cancer Cell Line Encyclopedia (CCLE) database (**Supplementary Fig. 2b**). Likewise, a positive correlation between CARM1 and CARM1/XBP1 target genes was also observed in the TCGA ovarian cancer datasets (**Fig. 2f**). Further, as requested, we now performed the experiments *in vivo* using a CARM null xenograft using both A1847 human cell lines and UPK10 mouse cell lines. Our results show that B-109 did not significantly reduce the burden of tumors formed by CARM1 null cells (**Supplementary Fig. 5b and 6b**).

What was the rationale for choosing PEO4 cell line over OVSAHO, which based on the western

blot appears to express a much higher level of CARM1 compared to PEO4? Including an explanation would be useful.

Response: We did not have a rationale to choose PEO4 over OVSAHO beyond the simple reason that when we first started this project, we have PEO4 in the lab already. Regardless, to completely address the reviewer the comments, we now generated OVSAHO CARM1 knockout cell line and show that same as in A1847 and PEO4 cells, XBP1s target gene expression and sensitivity to B-I09 was reduced by CARM1 knockout (**Supplementary Fig. 2g and 4d**).

Fig 5F: CARM1 band for OVSAHO runs below PEO4 & A1847. A comment would be helpful.

Response: We thank the reviewer for the comment on the gel appearance. This might be due to too many samples in the same gel, where the samples tend to have “smile effects”. Regardless, we repeated this experiment to show that the CARM1 band for OVSAHO does not typically run below those in other cell lines when only limited number of samples were used (Figure 3 for reviewer).

Minor comments

In addition to reference 23, in this manuscript providing a more detailed description of the PDX, such as the original patient characteristics from which the PDXs were derived, would be helpful to the reader.

Response: We thank the reviewer for the suggestion and have now included a more detailed description of the PDX as requested in the methods section on page 23, paragraph 2.

Line 123 “CAMR1” should be CARM1

Response: We thank the reviewer for spotting the typo that is now corrected.

Reviewer #3 (Remarks to the Author):

In this manuscript, Lin et al. present data suggesting that CARM1-expressing ovarian cancer cells are selectively sensitive to inhibition of the IRE1 α /XBP1s pathway, both in vitro and in vivo. They find that CARM1 interacts with XBP1s and co-regulates gene expression in ovarian cancer cells, some that are distinct. Furthermore, they show that CARM1 sensitizes ovarian cancer cells to an IRE1 α inhibitor (B-109), either alone or with a checkpoint inhibitor. However, molecular mechanism(s) of how this happens is not clear. This work is an extension of the authors' earlier work published in Nature Comm. showing that EZH2 inhibition is a potential strategy for to target CARM1 expressing ovarian cancer. The manuscript has significant data that support the role of CARM1 on ER stress signaling and in ovarian cancer. However, there are some significant issues and mechanistic details that are missing.

1. Throughout the text and figures, XBP1 designation is erroneously used for XBP1s; this has to be corrected.

Response: We thank the reviewer for pointing this out and we have now corrected this error accordingly.

2. In Figure 3, a larger panel of XBP1s-CARM1 target genes should be presented in more than one cell line, to show that what is observed in global analyses can be validated individually.

Response: As requested, we now validated several XBP1s/CARM1 target genes in control and CARM1 knockout isogenic A1847 and PEO4 cell lines (**Fig. 2h and Extended Fig. 2i**).

3. Does a gene expression signature of XBP1s-CARM1 regulated genes have utility as a prognostic marker for ovarian cancer?

Response: We explored the correlation between XBP1s/CARM1 target genes signature score and survival in the TCGA datasets. However, although as previously reported that CARM1 expression predicts shorter survival⁵, we did not observe a significant correlation between XBP1s/CARM1 regulated gene signature and survival (data not shown). This is consistent with the notion that regulation of ER stress response is highly context dependent by multiple mechanisms both cell intrinsically and extracellular microenvironment⁶⁻⁸.

4. The data presented on the physical interactions between CARM1 and XBP1s are weak (Figure 4a, Supplementary Figure 3). In Supplementary Figure 3A, IP-Western analysis is shown with the endogenous proteins in A1347 cells, the most robust way to interrogate potential interactions between two factors. The quality of the data in this figure is very low; one can barely discern an XBP1s band among the large background in the specific IP lane. Similarly, in the GST-pulldown assay presented in Supplementary Figure 3C, there are extremely weak bands, despite the large amounts of GST fusion proteins used. An unrelated GST fusion protein can be used in these experiments as a control.

Response: The challenge with the co-IP experiment has to do with the fact that the molecular weight of XBP1s is very close to that of heavy chain IgG. To address the reviewer's comments, we now performed new experiments using glycine buffer after co-IP analysis to reduce IgG heavy chain background (**Fig. 4a**). For the GST pulldown experiment, in addition to GST, GST-1-140 and GST481-608 were used as additional controls (**Supplementary Fig. 3b**).

5. In Figure 4A, there are two bands for XBP1s in the pull-down sample, whereas in the other

figures (e.g. Figure 2B) a single band is observed. The authors should provide an explanation for this.

Response: We and others always observe two very close bands for XBP1s^{4,9,10}. Whether this can be observed depends on conditions such as percentage of gels, how long one runs the gel and exposure during immunoblotting development etc. For example, in the pull-down experiments, we ran 10% gel that reveals two bands, while in other experiments we typically run 8% gels. To further address the reviewer's comments, we repeated the immunoblot experiments in Figure using 10% gel, which now reveals two bands (**Fig. 2b**).

6. A previous study from the same group (Karakashev et al., 2018) has shown that CARM1 knockout ovarian cells exhibit significant decrease in growth. Thus, the IC50 values presented in Figure 5 will not properly reflect the sensitivity of a cell line to a certain drug (Hafner et al., 2016, PMID:27135972). Alternative methods should be employed.

Response: We apologize for the confusion. As we published previously^{5,11}, we normalized our results to account for the difference in growth caused by CARM1 knockout in all our results. More importantly, our findings were validated in vivo in xenograft and PDXs models. Regardless, as requested, to validate our approach, we performed the suggested growth rate inhibition (GR) metrics analysis using the suggested methods. Consistently with our analysis, the results using GR metrics show that CARM1 knockout significantly decreased

response to B-109 (Figure 4 for reviewer).

7. In the same study (Karakashev et al., 2018) the authors have shown that CARM1 expressing cells are selectively sensitive to EZH2 inhibition. Is there synergy between EZH2 and IRE1 α inhibition in ovarian cancer cells?

Response: As requested, we now performed the combination experiment and our results show that there is no significant synergy between GSK126 and B-109 based on the combination index calculations (Figure 5 for reviewer).

8. The authors have generated CRISPRi-mediated CARM1 activated cells and tested their response to B-109 (Figure 5). Did they measure IRE1/XBP1s pathway activity in these cells? For example, did CARM1 activation enhance IRE1 α phosphorylation/activation, XBP1 splicing, and XBP1s transcriptional activity including

its target gene expression? This information is necessary to properly interpret the data presented.

Response: Our results show that CARM1 functions to control XBP1s' association with its target genes downstream of XBP1 splicing downstream of IRE1 α (**Fig. 1**). Regardless, our new results show that CARM1 activation promotes the expression of XBP1s target genes without affecting XBP1s expression (**Supplementary Fig. 2h and 4j**). In addition, CARM1 activation promotes XBP1s transcriptional activity as evidenced by an increase in its reporter activity (**Fig. 2j**).

9. The data provided suggest that CARM1 expressing cells may have enhanced XBP1s activation. Thus, CARM1 expression should correlate with XBP1s activity in patient material; is this so? This analysis can easily be done by scoring for expression of the XBP1s target gene signature and CARM1 expression. The authors should examine whether this correlation is present in multiple publicly available ovarian cancer gene expression datasets.

Response: As stated by the reviewer, we show that CARM1 expression positively correlates with XBP1s target genes in both the TCGA ovarian cancer datasets and the CCLE datasets (**Fig. 2f and Supplementary Fig. 2b**).

10. Supplementary Fig. 3e-f: Although inhibition of CARM1 enzymatic activity does not affect the expression of two select CARM1/XBP1 target genes, it cannot exclude the possibility that CARM1 inhibition may affect sensitivity of IRE1 to B-I09.

Response: As suggested, we assessed the effect of CARM1 inhibitor on IRE1 α with or without B-I09 treatment by examining the ratio between spliced XBP1 (XBP1s) and total XBP1 (XBP1t) mRNAs. Our results show that CARM1 inhibitor did not change the splicing efficiency in both control and B-I09 treated cells (**Supplementary Fig. 3f**), which supports that CARM1 inhibitor does not affect the sensitivity of IRE1 to B-I09.

11. The authors suggest a model whereby CARM1 determines ER stress response by controlling XBP1s association with its target genes. However, direct evidence of this was not provided. Re-ChIP experiments should be provided to interrogate this possibility.

Response: We thank the reviewer for the suggestion. We now performed the requested experiments and, as predicted, our Re-ChIP results further confirm our proposed model (**Fig. 3d**).

12. Does CARM1-dependent sensitivity to B-I09 rely on any of the CARM1/XBP1s targets identified? Are there other CARM1 downstream effectors (independent of the IRE1-XBP1s pathway) that are implicated in regulating ovarian cancer? A global gene expression experiment and its validation can answer these questions.

Response: As we and others have published, CARM1 is implicated in a number of pathways that are important for cancer. The focus of the present study is on its newly identified role in regulating XBP1s pathway. Thus, other pathways regulated by CARM1 is beyond the scope of the present studies. However, our new results show that BI-09 treatment significantly induced pro-apoptotic CHOP expression (**Supplementary Fig. 4k-l**). In addition, we show that knockdown of CHOP expression significantly reduced the sensitivity of BI-09 in CARM1-expressing cells, which correlates with a suppression of apoptosis induced by B-I09 (**Fig. 5h-i and Supplementary Fig. 4l-m**). Thus, these new data support that CHOP induction contributes

to sensitivity induced by B-109.

Reviewer #4 (Remarks to the Author):

Lin et al in this article performed CUT&RUN experiment to assay distribution of CARM1 and compared this to XBP1 binding and H3K27ac in A1847 cells with and without tunicamycin, an ER stress inducer.

I have several concerns related to the computational analyses of this paper (not in particular order of importance):

- Normalization is important to allow comparison between CUT&RUN experiments, and for conclusion such as in Fig 1g. How are authors normalizing the data? The authors say "default normalization parameters and call significant binding peaks for CARM1, XBP1 vs input control using options "-style factor". It is not clear what these normalization parameters are, and what approach is used.

Response: HOMER was used for normalization. Its default normalization parameter is to convert alignments to values that are the number of tags per 1bp per 10M reads. We now include the description in the methods section on page 25, paragraph 2.

- the overlap between DE genes from RNAseq and the CUT&RUN peak mapped genes is not very convincing (Fig 2c), mostly because of the way the analysis is currently done. In Fig 2c, there are 543 DE genes, 10956 XBP1 mapped genes, and 8340 CARM1 mapped genes. Given the number of XBP1 genes are so high (50% of genome), it has a close to 50% random chance to overlap with any gene set of interest. So it renders the overlap of 363 / 543 to be quite meaningless. A better way to analyze is to select equal number of genes from all three groups (i.e. select 500 top XBP1 peak mapped genes by intensity, same for CARM1 top 500 peak-mapped genes), and calculate 3-way overlap. Alternatively, you can lower threshold of DESeq2 to select more DE genes from RNAseq side, but keep each group same size.

- Related to Fig 2c, can authors sort CUT&RUN peak-mapped genes into bins by intensity (or Fold change over vehicle), and plot the number of DE genes that overlap with CUT&RUN genes in each bin? This provides a better assessment of the correlation between CUT&RUN and the targeted gene expression.

Response: Thank you for the nice suggestion! While the overlap of 363/543 is not the overlap between XBP1 mapped genes (10956) and RNA-seq genes (543), it is the overlap between RNA-seq (543 genes) and genes that occupied by both CARM1 and XBP1 (7421 genes). We calculated enrichment and significance of the overlap using Fisher Exact Test, using RNA-seq (543 genes), genes occupied by both CARM1 and XBP1 (7421 genes) and a pool of 21240 genes found to be expressed in our experiment. The calculation indicated that 363 of 543 genes is significant with 1.9-fold over the chance alone. We appreciate the suggestion and add the plot for the assessment of the correlation between CUT&RUN and the targeted gene expression using 500 top XBP1 peak mapped genes by fold change (**Supplementary Fig. 1c**). In this plot, we sorted CUT&RUN peak-mapped genes by XBP1 binding fold change over vehicle. Using a 500 gene window, 4 value were plotted: % of genes that are induced by Tunicamycin and decreased by both CARM1 KO and shXBP1, mean log₂ ratio of Tu/vehicle, mean log₂ (CARM1 KO/Ctrl), mean log₂ (shXBP1/Ctrl). All 4 trends indicate correlation of DNA binding with effect on expression. The genes bound by both XBP1 and CARM1 that have much more higher binding signal in Tu/Vehicle (red line) have more chance to be confirmed by RNA-seq to be a direct

target (purple line), and correlate with stronger decreased expression by CARM1 KO (green line) and shXBP1(yellow line).

- The statistical significance reported in some of the scatterplot heatmaps is at odds with the strength of the correlation (Fig 2e, f). Because you have large sample size, it is very easy to obtain statistical significance. In so doing, one maybe tricked into believing the results are very strong and meaningful, when in fact the correlation value is quite weak (Fig 2f, $R=0.225$, but with $p=7 \times 10^{-5}$). I suggest removing P-values in these plots, and just showing R and perhaps indicate N (sample size). What is the pearson correlation for these scatterplots? Fig 2e does not show a scatterplot that agrees with a $R=0.402$ - the points seem randomly distributed. For both Fig 2e, 2f, can authors switch to density plot (similar to Fig 1f)?

Response: As requested, we now switch to density plots for **Fig. 2e and 2f** and the corresponding pearson correlation values are $r = 0.207$ ($P = 3 \times 10^{-4}$) and $r = 0.386$ ($P = 2 \times 10^{-14}$), respectively. In addition, we removed the P values and indicated N (sample size) as requested.

- Have the authors performed Chip-seq of XBP1 and of CARM1 and compared with CUT&RUN to validate their CUT&RUN experiments? What is the advantage of CUT&RUN over Chip-seq for these two factors?

Response: We tried ChIP-seq for these factors with no success. This is the reason why we switched to CUT&RUN. Based on our experience, these assays are highly dependent on antibodies used. Some antibodies only work in one of these assays. Regardless, we validated our CUT&RUN findings using ChIP-qPCR analysis on selected XBP1s/CARM1 target genes (**Fig. 3**).

- The authors reported 22,398 CARM1 CUT&RUN peaks, and ?? XBP1 CUT&RUN peaks. I believe that care should be taken in general when interpreting peaks from CUT&RUN experiments, because of the effect of indirect binding (Skene et al, 2017, PMID: 28079019). Direct binding peaks are usually distinguished by the presence of a consensus motif and protection of motif-bound region from pA-MNase enzyme cut due to TF occupancy. Fortunately, direct binding peaks can be teased apart computationally by checking the frequency of cuts within the motif core region and compare with flanking region. This is known as motif footprinting analysis. Can authors perform motif footprinting analysis (Zhu et al, 2019 PMID: 31500663) (Neph et al, 2012, PMID: 22955618) (Pique-Regi et al, 2011, PMID: 21106904), to further confirm that the CUT&RUN peaks are direct binding, and comment the extent of indirect

binding? This is critical piece of information for establishing that CARM1 and XBP1 bind to target genes promoters. Since XBP1 has a known motif, it should be expected that targeted gene promoters with XBP1 motif should have no pA-MNase enzyme cuts. So for XBP1 and CARM1: check for the presence of footprints on XBP1 motif.

Response: As suggested, we ran the CUN&RUNTools (Zhu

et al, 2019 PMID: 31500663) for both CARM1 CUT&RUN. XBP1 motif is still among the top hit enriched by CARM1 CUT&RUN peaks (Figure 6A for reviewer), respectively. Enzyme cut protection is observed in motif core and deprotected in the flanking regions (Figure 6B for reviewer). Regardless, the importance of CARM1 in regulating XBP1 pathway was extensively validated both in vitro and in vivo in functional studies.

- RNAseq analyses missing very important volcano plot. It is not clear if the number of DE genes are derived using a combination of significance threshold and fold change values, or just significance value.

Response: DE genes were derived using only significance FDR values. Volcano plot for Tu/Vehicle genes with highlighted shXBP1 and CARM1 knockout genes was generated (**Supplementary Fig. 1b**).

- Methods on DEseq2 section of RNASeq analysis: "Overall gene expression changes were considered significant if passed FDR<5% thresholds unless stated otherwise." Authors may consider lowering FDR threshold to 10%. 5% maybe considered too stringent.

Response: Thank you for the suggestion. The reason we used FDR<5% because it already gave us substantial numbers of differentially expressed genes. By using FDR<5%, we got 3313 genes that significantly induced by tunicamycin, 6012 genes downregulated by CARM1, 1276 genes downregulated by shXBP1. We felt that using FDR<5% would result in more statistically sound results, with the most robust set of genes.

- Fig 1a, e are missing color scale bar.

Response: Thank you, we have added the color scale bars (**Fig. 1a and 1e**).

- Fig 2d color bars: can authors change the blue end of "fold increase" color scale to use a different color? The color currently overlaps with the log(1+reads/10M) color bar, which is also blue.

Response: Thank you. We have changed to different colors to avoid the same color scale between RNA and binding panels (**Fig. 2d**).

Cited references

1. Hafner, M., Niepel, M., Chung, M. & Sorger, P.K. Growth rate inhibition metrics correct for confounders in measuring sensitivity to cancer drugs. *Nat Methods* **13**, 521-527 (2016).
2. Condamine, T. *et al.* Lectin-type oxidized LDL receptor-1 distinguishes population of human polymorphonuclear myeloid-derived suppressor cells in cancer patients. *Sci Immunol* **1** (2016).
3. Tang, C.H. *et al.* Secretory IgM Exacerbates Tumor Progression by Inducing Accumulations of MDSCs in Mice. *Cancer Immunol Res* **6**, 696-710 (2018).
4. Tang, C.H. *et al.* Inhibition of ER stress-associated IRE-1/XBP-1 pathway reduces leukemic cell survival. *J Clin Invest* **124**, 2585-2598 (2014).
5. Karakashev, S. *et al.* CARM1-expressing ovarian cancer depends on the histone methyltransferase EZH2 activity. *Nature communications* **9**, 631 (2018).
6. Song, M. & Cubillos-Ruiz, J.R. Endoplasmic Reticulum Stress Responses in Intratumoral Immune Cells: Implications for Cancer Immunotherapy. *Trends Immunol* **40**, 128-141 (2019).
7. Cubillos-Ruiz, J.R., Bettigole, S.E. & Glimcher, L.H. Tumorigenic and Immunosuppressive Effects of Endoplasmic Reticulum Stress in Cancer. *Cell* **168**, 692-706 (2017).
8. Urra, H., Dufey, E., Avril, T., Chevet, E. & Hetz, C. Endoplasmic Reticulum Stress and the Hallmarks of Cancer. *Trends Cancer* **2**, 252-262 (2016).
9. Lee, A.H., Scapa, E.F., Cohen, D.E. & Glimcher, L.H. Regulation of hepatic lipogenesis by the transcription factor XBP1. *Science* **320**, 1492-1496 (2008).
10. Rodriguez, D.A. *et al.* BH3-only proteins are part of a regulatory network that control the sustained signalling of the unfolded protein response sensor IRE1alpha. *EMBO J* **31**, 2322-2335 (2012).
11. Karakashev, S. *et al.* EZH2 Inhibition Sensitizes CARM1-High, Homologous Recombination Proficient Ovarian Cancers to PARP Inhibition. *Cancer Cell* **37**, 157-167 e156 (2020).

REVIEWER COMMENTS

Reviewer #1 (Remarks to the Author):

The authors have satisfactorily addressed all of my comments. In light of the new data implicating proapoptotic CHOP induction upon IRE1 inhibition in CARM1^{hi} cells, it would be relevant to mention PMID: 31672843, which reports similar observations. Overall, the current manuscript by Lin et al is very interesting, mechanistically rich, and therapeutically relevant.

Reviewer #2 (Remarks to the Author):

The authors responded to the previous comments with additional data, explanation and interpretation. They were highly responsive and thus the manuscript continues to be very strong and rigorous. I am satisfied with the revised manuscript and commend the authors on this important and exciting study.

Reviewer #3 (Remarks to the Author):

In this manuscript, Lin et al showed that CARM1-expressing ovarian cancer cells are selectively sensitive to inhibition of the IRE1 α /XBP1s pathway, both in vitro and in vivo, with significant experimental evidence. Mechanistically, they found that CARM1 interacts and co-regulates gene expression with XBP1s in ovarian cancer cells. They also have identified a set of CARM1/XBP1s co-regulated genes. However, the exact mechanism of how CARM1 confers sensitivity to IRE1 α /XBP1s pathway inhibitor was not explored and thus some key findings are preliminary. For example, does CARM1-dependent sensitivity to B-I09 rely on any of the CARM1/XBP1s targets? Are other CARM1 downstream effectors (independent of XBP1s pathway) implicated in regulating cell sensitivity to B-I09?

Specific comments:

1. The authors have to use XBP1s (or XBP-1S etc.) for the spliced XBP1 protein/mRNA designation where needed in the text. At present, in a number of places only XBP1 is erroneously used to indicate XBP1s.
2. In Figure 4a, it is not clear which antibody was used for Western analysis. Since the band detected was only in the Tunicamycin-induced sample, it is likely that the anti-XBP1s antibody was used. In addition, two bands for XBP1s were detected in the pull-down sample. What are these two species? The authors should provide explanation for this.
3. In relation to Figure 5a, a previous study from the same group (Karakashev et al., 2018) has shown that CARM1 knockout cells exhibit remarkable decrease in growth. Under these conditions, the IC₅₀ value may not properly reflect the sensitivity of a cell line to a certain drug (Hafner et al., 2016. <https://pubmed.ncbi.nlm.nih.gov/27135972/>).
4. The authors have generated CARM1-activated cells and tested their response to B-I09. Did they measure IRE1/XBP1s pathway activity in these cells? Did CARM1 activation enhance XBP1s transcriptional activity and its target gene expression?
5. The results in this manuscript suggested that CARM1 expressing cells may have enhanced XBP1s activation. Thus CARM1 expression levels should correlate with XBP1s activity (e.g. can easily be scored using XBP1s target gene expression). The authors should examine whether this correlation is present by testing multiple ovarian cancer gene expression datasets.
6. Extended Data Fig. 3e-f: Although inhibition of CARM1 enzymatic activity does not affect the expression of CARM1/XBP1 target genes, it cannot exclude the possibility that CARM1 inhibitor does not affect cell sensitivity to B-I09.

Reviewer #4 (Remarks to the Author):

The manuscript is much improved using the suggestions I have proposed. Here are my comments.

1. Regarding Supplementary Figure 1, the authors can consider adding the volcano plot, panel c,

to the main figure. For c, can authors add another line plotting the fold-over-random for the overlap at each of 500 gene window? This will allow readers better assess the significance of overlap at the top of the list, without choosing a hard cutoff on number of genes.

2. Regarding the cut and run motif footprinting analysis (Fig 6 for reviewer), the analysis indicates the enrichment of XBP1 motif, but it is not ranked at the top (it is at #8). What might be the reason for this? How many cut and run peaks have this motif? In (b), it appears that the motif footprinting analysis was done not properly, because the plot is not symmetrical around the center (distance=0). It seems that there is protection of DNA cuts, but because of assymetry, it is hard to assess. Most likely my suspicion is that motif footprinting analysis was not done correctly. Can authors fix this problem, and attach the results to the paper? It would serve important validation purpose of cut and run.

A point-by-point response to the reviewers' comments

We thank the reviewers for their comments. As instructed by the Editor, we now further clarified points #1, #3 and #4 raised by Reviewer #3 and those raised by Reviewer #4. In particular as specified by the Editor, we provided experimental data showing that B-I09 suppresses IRE1/XBP1s activity equally efficiently in CARM1 low cells with or without endogenous CARM1 upregulation.

REVIEWER COMMENTS

Reviewer #1 (Remarks to the Author):

The authors have satisfactorily addressed all of my comments. In light of the new data implicating proapoptotic CHOP induction upon IRE1 inhibition in CARM1^{hi} cells, it would be relevant to mention PMID: 31672843, which reports similar observations. Overall, the current manuscript by Lin et al is very interesting, mechanistically rich, and therapeutically relevant.

Response: We thank the review for the positive comments. We have now included the reference in the revised manuscript as requested (reference #22).

Reviewer #2 (Remarks to the Author):

The authors responded to the previous comments with additional data, explanation and interpretation. They were highly responsive and thus the manuscript continues to be very strong and rigorous. I am satisfied with the revised manuscript and commend the authors on this important and exciting study.

Response: We thank the reviewer for the positive comments.

Reviewer #3 (Remarks to the Author):

In this manuscript, Lin et al showed that CARM1-expressing ovarian cancer cells are selectively sensitive to inhibition of the IRE1 α /XBP1s pathway, both in vitro and in vivo, with significant experimental evidence. Mechanistically, they found that CARM1 interacts and co-regulates gene expression with XBP1s in ovarian cancer cells. They also have identified a set of CARM1/XBP1s co-regulated genes. However, the exact mechanism of how CARM1 confers sensitivity to IRE1 α /XBP1s pathway inhibitor was not explored and thus some key findings are preliminary. For example, does CARM1-dependent sensitivity to B-I09 rely on any of the CARM1/XBP1s targets? Are other CARM1 downstream effectors (independent of XBP1s pathway) implicated in regulating cell sensitivity to B-I09?

Response: As we and others have published, CARM1 is implicated in a number of pathways that are important for cancer. The focus of the present study is on its newly identified role in regulating XBP1s pathway. Thus, other pathways regulated by CARM1 is beyond the scope of the present studies. However, our new results show that B-I09 treatment significantly induced pro-apoptotic CHOP expression (**Supplementary Fig. 4m-n**). In addition, we show that knockdown of CHOP expression significantly reduced the sensitivity of B-I09 in CARM1-expressing cells, which correlates with a suppression of apoptosis induced by B-I09 (**Fig. 5h-i and Supplementary Fig. 4o**). Thus, these new data support that CHOP induction contributes to sensitivity induced by B-I09.

Specific comments:

1. The authors have to use *XBP1s* (or *XBP-1S* etc.) for the spliced *XBP1* protein/mRNA designation where needed in the text. At present, in a number of places only *XBP1* is erroneously used to indicate *XBP1s*.

Response: We now went through the designation to make sure that we used *XBP1s* with two exceptions whereby either shRNAs against *XBP1* was used because they target both unspliced and spliced *XBP1* or anti-*XBP1* antibody was used because the antibody recognizes both unspliced and spliced *XBP1*.

2. In Figure 4a, it is not clear which antibody was used for Western analysis. Since the band detected was only in the Tunicamycin-induced sample, it is likely that the anti-*XBP1s* antibody was used. In addition, two bands for *XBP1s* were detected in the pull-down sample. What are these two species? The authors should provide explanation for this.

Response: The antibody used in Figure 4a recognizes both unspliced *XBP1* and spliced *XBP1s*, which differs by molecular weight due to IRE1 RNase activity caused frameshift. The reviewer is correct that the bands detected here are *XBP1s* based on their molecular weights. This is consistent with the notion that *XBP1* is rapidly degraded by proteasome and typically undetectable unless proteasome activity is inhibited^{2,3}. We and others always observe two very close bands for *XBP1s*⁴⁻⁶. This is due to post-translational modifications occurred on *XBP1s*⁷.

3. In relation to Figure 5a, a previous study from the same group (Karakashev et al., 2018) has shown that *CARM1* knockout cells exhibit remarkable decrease in growth. Under these conditions, the IC₅₀ value may not properly reflect the sensitivity of a cell line to a certain drug (Hafner et al., 2016. <https://pubmed.ncbi.nlm.nih.gov/27135972/>).

Fig.4 for reviewer: *CARM1* knockout decreases sensitivity to B-109 determined by GR metrics analysis. Calculation of GR values using endpoint drug response data¹.

Response: As we published previously^{8,9}, we normalized our results to account for the difference in growth caused by *CARM1* knockout in all our results. More importantly, our findings were validated in vivo in xenograft and PDXs models. Regardless, as requested, to validate our approach, we performed the suggested growth rate inhibition (GR) metrics analysis using the suggested methods. Consistently with our analysis, the results using GR metrics show that *CARM1* knockout significantly decreased response to B-109 (Figure 1 for reviewer).

4. The authors have generated *CARM1*-activated cells and tested their response to B-109. Did they measure IRE1/*XBP1s* pathway activity in these cells? Did *CARM1* activation enhance *XBP1s* transcriptional activity and its target gene expression?

Response: We assessed IRE1 activity upon B-109 treatment in these cells by using the ratio of spliced *XBP1s* and unspliced total *XBP1t* as a readout (*XBP1s*/*XBP1t*). Our new results show that *CARM1* expression does not affect IRE1 enzymatic activity and B-109 inhibits IRE1 enzymatic activity equally efficiently in these cells (**Supplementary Fig. 4e**). Moreover, *CARM1*

inhibitor did not affect the sensitivity to B-I09 (**Supplementary Fig. 4i**). Finally, we showed that CARM1 activation enhanced XBP1s transcriptional activity (**Fig. 2k**) and its target gene expression (**Supplementary Fig. 2h**). Together, these results are consistent with our findings that CARM1 functions as a co-activator of XBP1s to promote XBP1s target gene expression downstream of IRE1.

5. The results in this manuscript suggested that CARM1 expressing cells may have enhanced XBP1s activation. Thus CARM1 expression levels should correlate with XBP1s activity (e.g. can easily be scored using XBP1s target gene expression). The authors should examine whether this correlation is present by testing multiple ovarian cancer gene expression datasets.

Response: Our data show that CARM1 expression positively correlates with XBP1s target genes in both the TCGA ovarian cancer datasets and the CCLE datasets (**Fig. 2h and Supplementary Fig. 2b**).

6. Extended Data Fig. 3e-f: Although inhibition of CARM1 enzymatic activity does not affect the expression of CARM1/XBP1 target genes, it cannot exclude the possibility that CARM1 inhibitor does not affect cell sensitivity to B-I09.

Response: As requested, we now show that CARM1 inhibitor does not affect cell sensitivity to B-I09 (**Supplementary Fig. 4i**).

Reviewer #4 (Remarks to the Author):

The manuscript is much improved using the suggestions I have proposed. Here are my comments.

1. Regarding Supplementary Figure 1, the authors can consider adding the volcano plot, panel c, to the main figure. For c, can authors add another line plotting the fold-over-random for the overlap at each of 500 gene window? This will allow readers better assess the significance of overlap at the top of the list, without choosing a hard cutoff on number of genes.

Response: As requested, we now adding the volcano plot and panel to the main Figure 2. In addition, as suggested, we added another line plotting the fold-over-random for the overlap at each of 500 gene window in the **new Fig. 2f**.

2. Regarding the cut and run motif footprinting analysis (Fig 6 for reviewer), the analysis indicates the enrichment of XBP1 motif, but it is not ranked at the top (it is at #8). What might be the reason for this? How many cut and run peaks have this motif? In (b), it appears that the motif footprinting analysis was done not properly, because the plot is not symmetrical around the center (distance=0). It seems that there is protection of DNA cuts, but because of assymetry, it is hard to assess. Most likely my suspicion is that motif footprinting analysis was not done correctly. Can authors fix this problem, and attach the results to the paper? It would serve important validation purpose of cut and run.

Response: We thank the reviewer for the insightful comments. The reviewer is correct that the analysis was not done properly due to the fact the pipeline from the paper the reviewer suggested was designed for paired-end raw reads input, while our ChIP-seq is single-end. Accordingly, we now re-ran the analysis with single-end setting. With 6,543 peaks with XBP1 motif were assayed to find positions of exact DNA cuts (**Supplementary Fig. 1b**). We observed

a symmetric plot demonstrating protection of motif location from DNA cuts. In addition, our new results reveal that XBP1s motif is now ranked #3 in the analysis. Notably, there is no specific binding protein for the #1 ranked motif that is not centrally enriched, while NFY binding motif ranked #2. Our results are thus consistent with previous studies whereby NFY binding motif ranks among the top motifs in XBP1s ChIP-seq analysis ¹⁰.

Cited References

- 1 Hafner, M., Niepel, M., Chung, M. & Sorger, P. K. Growth rate inhibition metrics correct for confounders in measuring sensitivity to cancer drugs. *Nat Methods* **13**, 521-527, doi:10.1038/nmeth.3853 (2016).
- 2 Calfon, M. *et al.* IRE1 couples endoplasmic reticulum load to secretory capacity by processing the XBP-1 mRNA. *Nature* **415**, 92-96, doi:10.1038/415092a (2002).
- 3 Yoshida, H., Matsui, T., Yamamoto, A., Okada, T. & Mori, K. XBP1 mRNA is induced by ATF6 and spliced by IRE1 in response to ER stress to produce a highly active transcription factor. *Cell* **107**, 881-891, doi:10.1016/s0092-8674(01)00611-0 (2001).
- 4 Lee, A. H., Scapa, E. F., Cohen, D. E. & Glimcher, L. H. Regulation of hepatic lipogenesis by the transcription factor XBP1. *Science* **320**, 1492-1496, doi:10.1126/science.1158042 (2008).
- 5 Rodriguez, D. A. *et al.* BH3-only proteins are part of a regulatory network that control the sustained signalling of the unfolded protein response sensor IRE1alpha. *EMBO J* **31**, 2322-2335, doi:10.1038/emboj.2012.84 (2012).
- 6 Tang, C. H. *et al.* Inhibition of ER stress-associated IRE-1/XBP-1 pathway reduces leukemic cell survival. *J Clin Invest* **124**, 2585-2598, doi:10.1172/JCI73448 (2014).
- 7 Wang, F. M., Chen, Y. J. & Ouyang, H. J. Regulation of unfolded protein response modulator XBP1s by acetylation and deacetylation. *Biochem J* **433**, 245-252, doi:10.1042/BJ20101293 (2011).
- 8 Karakashev, S. *et al.* EZH2 Inhibition Sensitizes CARM1-High, Homologous Recombination Proficient Ovarian Cancers to PARP Inhibition. *Cancer Cell* **37**, 157-167 e156, doi:10.1016/j.ccell.2019.12.015 (2020).
- 9 Karakashev, S. *et al.* CARM1-expressing ovarian cancer depends on the histone methyltransferase EZH2 activity. *Nature communications* **9**, 631, doi:10.1038/s41467-018-03031-3 (2018).
- 10 Pramanik, J. *et al.* Genome-wide analyses reveal the IRE1a-XBP1 pathway promotes T helper cell differentiation by resolving secretory stress and accelerating proliferation. *Genome Med* **10**, 76, doi:10.1186/s13073-018-0589-3 (2018).

REVIEWERS' COMMENTS

Reviewer #3 (Remarks to the Author):

Lin et al. responded to my previous comments in a satisfactory manner. I would like to suggest that the growth rate inhibition metrics analysis (presented as Fig. 1 for reviewer) data are presented as a Suppl. figure in the manuscript as it clarifies a critical point, per my previous comment.

Reviewer #4 (Remarks to the Author):

Thanks for the revision.

Regarding my concern 2, I am confused about "pipeline from the paper the reviewer suggested was designed for paired-end raw reads input, while our ChIP-seq is single-end". I asked about doing motif footprinting for *CUT&RUN* not for Chip-seq data.

Second, to my knowledge all CUT&RUN should be sequenced using paired-end sequencing. I don't know any motif footprinting analysis tools that can accept single-end reads as inputs. Could authors please clarify and let me know how motif footprinting was done? The figure that authors attached Supplementary Figure 1 shows + strand and - strand which leads me to believe that authors do have paired-end information (where + strand corresponds to 5' end of fragment, and - strand corresponds to 3' end of fragment). I suspect that the motif footprinting analysis may still have issues.

A point-by-point response to the reviewers' comments

Reviewer #3 (Remarks to the Author):

Lin et al. responded to my previous comments in a satisfactory manner. I would like to suggest that the growth rate inhibition metrics analysis (presented as Fig. 1 for reviewer) data are presented as a Suppl. figure in the manuscript as it clarifies a critical point, per my previous comment.

Response: As requested, the data is now included in the manuscript as **Supplementary Fig. 4a-b**.

Reviewer #4 (Remarks to the Author):

Thanks for the revision.

*Regarding my concern 2, I am confused about "pipeline from the paper the reviewer suggested was designed for paired-end raw reads input, while our ChIP-seq is single-end". I asked about doing motif footprinting for *CUT&RUN* not for Chip-seq data.*

Second, to my knowledge all CUT&RUN should be sequenced using paired-end sequencing. I don't know any motif footprinting analysis tools that can accept single-end reads as inputs.

Could authors please clarify and let me know how motif footprinting was done? The figure that authors attached Supplementary Figure 1 shows + strand and - strand which leads me to believe that authors do have paired-end information (where + strand corresponds to 5' end of fragment, and - strand corresponds to 3' end of fragment). I suspect that the motif footprinting analysis may still have issues.

Response: We apologize for the confusion. It should be "CUT&RUN" instead of ChIP-seq data. Upon further consultation with the reviewer and as requested by the editorial team, we now removed the motif footprinting figures from the manuscript since we only did single-end sequencing.